# Topology-Aware Learning of Tubular Manifolds via SE(3)-Equivariant Network on Ball B-Spline Curve

**Jingxuan Wang**[1][*]  **Zhongke Wu**[1]  **Xingce Wang**[1][†]  **Zeyao Zhang**[1]
**Chunghao Zheng**[1]  **Di Wang**[2]
[1]School of Artificial Intelligence, Beijing Normal University, China
[2]LILY Research Centre, Nanyang Technological University, Singapore

## Abstract

Tubular-like system shape analysis is quite difficult in geometry and topology, while it is widely used in plants and organs analysis in practice. However, traditional discrete representations such as voxels and point clouds often require substantial storage and may lead to the loss of fine-grained geometric and topological details. To address these challenges, we propose SE(3)-BBSCformerGCN, a novel framework for learning shape-aware representations from continuous tubular topological manifolds with equivariance to rotations and translations. Our approach leverages Ball B-Spline Curve (BBSC) to define tubular manifolds and its functional space. We provide a formal mathematical definition and analysis of the resulting manifolds and the BBSC functional space, and incorporate an equivariant mapping that preserves geometric and topological stability. Compared to the point cloud and voxel based representations, our manifold-based formulation significantly reduces data complexity while preserving geometric attributes together with topological features. We validate our method on the branch classification task for Circle of Willis (CoW) on the TopCoW 2024 dataset and the clinical dataset. Our method consistently outperforms voxel and point cloud based baselines in terms of classification performance, generalization ability, convergence speed, and robustness to overfitting.

## 1 Introduction

Tubular structures play a vital role in a wide range of tasks across computer vision, computer graphics, and interdisciplinary domains, including vascular segmentation [1], classification [2], and traffic network planning [3, 4]. However, the analysis of tubular-like models is quite difficult in both geometry and topology. From a geometric perspective, the branch morphology within tubular-like models varies significantly, with marked differences in tortuosity among individuals for branches of the same designation. From a topological perspective, the morphology and connectivity of the branches are not unique. Certain branches are prone to stenosis, absence, or duplication, and tubular-like model structures frequently exhibit fusion or crossover. Meanwhile, both the geometric structure and topological configuration of tubular-like models, such as vascular networks, may vary across different subjects or even between different scans of the same subject.

The B-Spline model [5, 6, 7], along with its extensions such as Ball B-Spline Curves (BBSC) [8], serves as an industry-standard tool widely adopted in computer graphics and computer-aided design (CAD). In this work, we propose a compact and differentiable representation for tubular structures based on BBSC, using a small set of control points, radii, and knot vectors. Compared to existing voxel-based methods [9] and point cloud-based methods [10], our model not only reduces storage

---

[*]First author. Email: 202431081034@mail.bnu.edu.cn
[†]Corresponding author. Email: wangxingce@bnu.edu.cn

overhead, but also enables accurate and interpretable geometric encoding. To enhance interpretability, we further define a smooth manifold structure over BBSC. To mitigate the sensitivity of geometric feature extraction to rigid transformations, we adopt a group-equivariant approach [11] and propose SE(3)-BBSCformer, a model that operates directly on the BBSC manifold and is equivariant under rotations and translations. By incorporating SE(3)-equivariance[12], our method ensures consistent feature extraction under arbitrary rotations and translations, leading to improved stability, accuracy, and generalization in tubular structure analysis.

To model the inter-branch topological connectivity, we use Graph Convolutional Networks (GCNs) [13] as the predominant approach for learning topological features. Building on this foundation, we innovatively propose a tubular manifold representation based on BBSC and construct a topology-aware SE(3)-equivariant network SE(3)-BBSCformerGCN on this manifold. Our main contributions are as follows: 1) We define a compact and interpretable tubular manifold based on BBSC, construct its associated functional space $\mathcal{M}$, and propose a metric on $\mathcal{M}$. An SE(3)-equivariant mapping, which enhances geometric consistency and symmetry preservation, is constructed from $\mathcal{M}$ to a higher-dimensional functional space $\mathcal{S}$, forming the architecture of SE(3)-BBSCformer. 2) We define a topologically glued structure composed of multiple BBSC manifolds and design a corresponding GCN to model the topological relationships between manifolds, which plays a critical role in handling structures with significant topological heterogeneity. 3) We conduct experiments on a multi-class classification task of the Circle of Willis (CoW) [14, 15]. Our proposed SE(3)-BBSCformerGCN demonstrates superior performance against voxel and point cloud state-of-the-art (SOTA) baselines—in terms of accuracy, computational efficiency, and generalization ability—on both a public dataset and a clinical dataset collected in collaboration with a medical institution.

## 2   Related Work

**Tubular structure extraction.** Deep learning has been widely applied to tubular structure analysis [16] . DSCNet [17] captures local vessel tortuosity via dynamic snake convolution, while DUNet [18] integrates deformable convolutions into UNet [19] for retinal vessel segmentation. DDT [20] employs a distance transform to enable geometry-aware voxel segmentation. Topological constraints have also been proven essential. GraphMorph [21] and TopoLab [22] explicitly model topological connections to improve performance on complex tubular structures. Meanwhile, TaG-Net [23] combines PointNet [24] with GNN [25] to extract both geometric and topological features for more accurate classification. The shape graph [26] models curves as edges and their intersections as nodes, enabling mathematical analysis of graph shape variations. Moreover, it introduces a multi-scale representation to simplify the expression of complex graphs. However, all these methods rely on discrete representations—pixels, voxels, or point clouds. In contrast, we propose a differentiable and compact representation using BBSC, enabling tubular structures to be modeled as manifolds with faithful geometric and topological expressiveness. Specifically, we represent tubular branches as BBSC-based nodes, while the connections between branches are treated as edges. This design avoids the complex process of defining edge weights and instead focuses on capturing the geometric characteristics of individual branches.

**Group equivariant network.** Group-equivariant models [11, 12] have become central in geometric deep learning [27] due to their robustness and interpretability. Group-equivariant Convolutional Networks (G-CNNs) [28] first introduced group symmetry into CNNs, showing strong performance on symmetry-rich imaging tasks. This idea has since been extended to 3D shape feature extraction, where models like Tensor Field Networks (TFN) [29], SE(3)-Transformer [30] incorporate SE(3)-equivariance into point cloud processing, significantly boosting both performance and stability. EquiTrack [31] achieves equivariance via steerable CNNs to process temporal sequences. SpaER [32] propose the construction of equivariant spatial means using steerable CNN filters and introduced an innovative use of self-attention mechanisms to process temporal sequences. MPerformer [33] employs an SE(3)-Transformer for molecular perception. However, the potential of applying group-equivariant models for geometric feature extraction in tubular structures has not been widely explored. To the best of our knowledge, we are the first to construct a topology-aware and group-equivariant network on manifolds and apply it to the shape analysis and feature extraction of tubular structures.

# 3 Preliminaries

**Ball B-Spline Curves (BBSCs).** BBSC is an extension of B-Spline curves to spheres, with each sphere defined by a control radius. This formulation provides a smooth representation for tubular objects with varying thickness. Given a set of control points $\{P_1, P_2, \ldots, P_n\}$, control radii $\{r_1, r_2, \ldots, r_n\}$, and a set of basis functions $N(t)$ which are computed by a knot vector $\{t_1, t_2, \ldots, t_{n+p+1}\}$, a BBSC of degree $p$ is defined as follows:

$$BBSC(t) = \sum_{i=0}^{n} N_{i,p}(t)C_i = \sum_{i=0}^{n} N_{i,p}(t)\,(P_i\,;\,r_i) = \left( \sum_{i=0}^{n} N_{i,p}(t)P_i\,;\,\sum_{i=0}^{n} N_{i,p}(t)r_i \right). \quad (1)$$

The basis function of degree 0 is initialized as $N_{i,0} = \begin{cases} 1, & \text{if } t_i \leq t < t_{i+1}, \\ 0, & \text{otherwise.} \end{cases}$, and the higher degree basis function $N_{i,p}(t)$ can be calculated by iteratively interpolating the knot vector $t_i, \ldots, t_{i+p+1}$ and $N_{i,p-1}(t)$, $N_{i+1,p-1}(t)$ as follows:

$$N_{i,p}(t) = \frac{t - t_i}{t_{i+p} - t_i} N_{i,p-1}(t) + \frac{t_{i+p+1} - t}{t_{i+p+1} - t_{i+1}} N_{i+1,p-1}(t). \quad (2)$$

A BBSC consists of two parts: the center B-Spline curve $\gamma(t) = \sum_{i=0}^{n} N_{i,p}(t)P_i$ and the radius function $\sigma(t) = \sum_{i=0}^{n} N_{i,p}(t)r_i$. Because $\gamma(t)$ and $\sigma(t)$ are calculated using the same basis function (Equation (1) and Figure 1), they inherently inherit the mathematical properties of B-Spline curves, including differentiability, local control, the convex hull property, and favorable topological characteristics. More importantly, compared to voxels and point clouds, the BBSC representation can continuously and differentiably represent a tubular object with a compact set of parameters, which not only reduces storage and computational complexity, but also enhances theoretical interpretability and enables accurate geometric and analytical modeling.

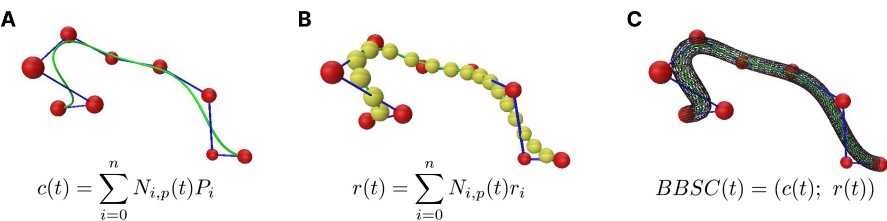

**A**    $c(t) = \sum_{i=0}^{n} N_{i,p}(t)P_i$    **B**    $r(t) = \sum_{i=0}^{n} N_{i,p}(t)r_i$    **C**    $BBSC(t) = (c(t);\ r(t))$

Figure 1: An example of BBSC. **A.** The 3D B-Spline curve is represented as a linear combination of control points (red balls) and their associated basis functions. **B.** The tubular surface is formed by sweeping balls (yellow balls) parameterized by control radii along the B-Spline curve. **C.** The BBSC providing a smooth and flexible curve representation in 3D Euclidean space.

**Group Representation**. Group [34] is a concept from abstract algebra [35] used to describe symmetries and related properties. A group $G$ consists of a set equipped with a binary operation $\circ$ and satisfies the following properties: closure, associativity and the existence of an identity element and an inverse element. The representation of a group $G$ is a homomorphic mapping [36] $\rho$: $G \mapsto GL(\mathcal{V}), \mathcal{V} \subseteq \mathbb{R}^D$, which satisfies $\forall g_1, g_2 \in G, \rho(g_1 \circ g_2) = \rho(g_1)\,\rho(g_2)$. $GL(\mathcal{V})$ denotes the set of all invertible linear transformations on $\mathcal{V}$ and $\rho(g)$ corresponds to the matrix representation of group element $g$. The abstract group operation can be realized as the corresponding matrix multiplication.

**Group Equivariance**. A function $\varphi : \mathcal{X} \to \mathcal{Y}$ is called $G$-equivariant if $\forall g \in G$, given a set of transformation $\rho_{\mathcal{X}}(g) : \mathcal{X} \to \mathcal{X}$, $\exists$ a transformation $\rho_{\mathcal{Y}}(g) : \mathcal{Y} \to \mathcal{Y}$, such that

$$\varphi(\rho_{\mathcal{X}}(g)x) = \rho_{\mathcal{Y}}(g)\varphi(x), \quad x \in \mathcal{X}. \quad (3)$$

Equation (3) can be simplified as $\varphi(\rho_{\mathcal{X}}(g)x) = \varphi(x)$ when the matrix representation of $\rho_{\mathcal{Y}}(g)$ is an identity matrix $I$. In this case, $\varphi$ is called invariant. If the function $\varphi$ is defined in a 3D Euclidean space $\mathbb{R}^3$ and it is equivariant under both rotation and translation transformations in this space, the function is referred to as 3D roto-translation equivariant (SE(3)-equivariant), the group is thus denoted as SE(3).

**SE(3)-Transformer**. The SE(3)-Transformer [30] is a deep learning-based method that applies group-equivariance to auxiliary geometric features on 3D point clouds. It constructs a set of SE(3)-equivariant basis kernels $W^{c_o c_i} \in \mathbb{R}^{(2c_o+1)\times(2c_i+1)}$ using spherical harmonics [37], corresponding Clebsch-Gordan coefficients [38], and a learnable radial function. These are then integrated with an attention mechanism [39] to form a group-equivariant network (see details in Appendix B.2). Given an input $f_{in}^{c_i}$ with $c_i$ channels, the output $f_{out}^{c_o}$ with $c_o$ channels is obtained as follows:

$$f_{out,i}^{c_o} = W_V^{c_o c_o} f_{in,i}^{c_o} + \sum_k \sum_{j \in \mathcal{N}_i} \alpha_{i,j} W_V^{c_o c_i}(x_i - x_j) f_{in,j}^{c_i}, \tag{4}$$

where the attention matrix is computed as follows:

$$\vec{q}_i = \bigoplus_{c_o} \sum_{c_i} W_Q^{c_o c_i} f_{in,i}^{c_i}, \; \vec{k}_{i,j} = \bigoplus_{c_o} \sum_{c_i} W_K^{c_o c_i}(x_i - x_j) f_{in,j}^{c_i}, \; \alpha_{ij} = \frac{\exp(\vec{q}_i^T \vec{k}_{i,j})}{\sum_{j' \in \mathcal{N}_i} \exp(\vec{q}_i^T \vec{k}_{i,j})}. \tag{5}$$

The direct sum $\bigoplus$ represents the concatenation of vectors. The attention mechanism is proven to be SE(3)-equivariant [30], as it relies on the inner product of $\alpha_{ij}$ and $W_V^{c_o c_i}(x_i - x_j)$, which is an SE(3)-equivariant operation.

## 4 Method and Analysis

Our method consists of two main components. First, we construct smooth BBSCs from discrete 3D data and, from a manifold perspective, define and analyze both their properties and the space they form. Second, we extend the SE(3)-Transformer to this manifold, yielding the SE(3)-BBSCformer for learning on continuous structures. We further employ a GCN to capture topological relations and design a pipeline (Figure 2) for branch classification in the CoW, a tubular arterial network with specific topological connections.

### 4.1 BBSC Functional Space and SE(3) Equivariant Mapping

**Definition 1**. We define the $p$-degree BBSC functional space $\mathcal{M}_p$ with given $P$, $r$, $\tau$ from their respective vector spaces as follows:

$$\mathcal{M}_p = \{\beta(P, r, \tau) \mid (P, r, \tau) \in \Theta\}, \quad \Theta = \{P \in \mathbb{R}^{3 \times n}, r \in \mathbb{R}_{>0}, \tau \in \mathbb{T}\}. \tag{6}$$

where $\mathbb{T} = \{t_i \in [0,1] | t_1 \leq t_2 \leq \cdots \leq t_{n+p+1}\} \in [0,1]^{n+p+1}$, and $\beta(P, r, \tau)$ is the total structure of BBSC (see Equation (1)), which is a two-dimensional manifold. In order to further stabilize BBSC, we impose certain constraints on $\mathbb{T} = \{t_i \in [0,1] | t_{p+1} < t_{p+2} < \cdots < t_n, t_1 = \cdots = t_{p+1} = 0, t_n = \cdots = t_{n+p+1} = 1\}$. The first and last $p + 1$ entries of the knot vector are set to 0 and 1, respectively, to ensure that the two endpoints of the curve coincide with the first and last control points. The parameter $t$ in Equation (1) specifies positions along the curve but does not affect the overall BBSC shape, serving only as an internal parameter independent of $\beta \in \mathcal{M}_p$. Because $\beta(P, r, \tau)$ is a linear combination of $P$ and $r$, it has infinitely many continuous partial derivatives. By excluding the first and last $p + 1$ entries of $\tau$, i.e., in the open subset of $\mathcal{M}_p$ that omits two boundary points, $N_{i,p}(t)$ becomes a recursive fractional linear function of $\tau$, and is differentiable with respect to $\tau$ of arbitrary order. Moreover, under the constraint that the middle entries of $\tau$ are strictly monotonically increasing, $\beta(P, r, \tau)$ possesses infinitely many continuous partial derivatives. According to Equation (1), $P$, $r$, and $\tau$ are in a separable product form. Consequently, their higher-order mixed partial derivatives can be expressed as products of higher-order partial derivatives, which remain continuous. Therefore, $\mathcal{M}_p$ can be regarded as $C^\infty(\Theta, \mathcal{M}_p)$. The complete proof is provided in Appendix C.

The metric on $\mathcal{M}_p$ can be simply defined as a weighted sum of the L2 norms of the parameters as follows:

$$g = \|P - \tilde{P}\|_2 + \alpha \|r - \tilde{r}\|_2 + \eta \|\tau - \tilde{\tau}\|_2. \tag{7}$$

Although the metric based on L2 norms provides a simple way to measure parameter variations, it ignores the perturbations on $\mathcal{M}_p$. Moreover, $\gamma(t)$ and $\sigma(t)$, which determine the BBSC shape, are independent (hence $P$ and $r$ are independent), whereas $\tau$ simultaneously influences both. Thus, the coupling between $\tau$ and $P$, $r$ should be considered when defining the metric. To address this, we

propose a metric that accounts for perturbations on $\mathcal{M}_p$ and parameter coupling. Concretely, we first characterize perturbations in the tangent space of $\mathcal{M}_p$ as $\delta\gamma$ and $\delta\sigma$ as follows:

$$\delta\gamma = \sum_{i=1}^n N_{i,p}(t)\delta P_i + \sum_{i=1}^n P_i\frac{\partial N_{i,p}(t)}{\partial\tau}\delta\tau, \ \delta\sigma = \sum_{i=1}^n N_{i,p}(t)\delta r_i + \sum_{i=1}^n r_i\frac{\partial N_{i,p}(t)}{\partial\tau}\delta\tau. \quad (8)$$

Given geometric perturbations $\delta\beta = (\delta\gamma, \delta\sigma)$ and $\tilde{\delta}\beta = (\tilde{\delta}\gamma, \tilde{\delta}\sigma)$ in the tangent space of $\mathcal{M}_p$, we define the following inner product as follows:

$$\langle\delta\beta, \tilde{\delta}\beta\rangle_{\mathcal{M}_p} = \int_0^1 \left( \langle\delta\gamma, \tilde{\delta}\gamma\rangle + \eta_\gamma\langle\frac{\partial\delta\gamma}{\partial t}, \frac{\partial\tilde{\delta}\gamma}{\partial t}\rangle + \alpha\delta\sigma\tilde{\delta}\sigma + \alpha\eta_\sigma\frac{\partial\delta\sigma}{\partial t}\frac{\partial\tilde{\delta}\sigma}{\partial t} \right)\omega(t)dt, \quad (9)$$

where $\alpha \geq 0$, $\eta_\gamma, \eta_\sigma \geq 0$, and $\omega(t)$ denotes a weighting function. One may simply set $\omega(t) = 1$ to perform unweighted integration. However, we recommend using $\omega(t) = \|\gamma'(t)\|$, which corresponds to arc-length weighting, thereby mitigating the effect of reparameterization and enhancing geometric interpretability. The proposed inner product is inspired by the Sobolev $H^1$ inner product [40, 41], which incorporates not only the Euclidean inner product of $(\delta\gamma, \delta\sigma)$ and $(\tilde{\delta}\gamma, \tilde{\delta}\sigma)$ themselves, but also their first-order derivatives. Because $\langle\cdot, \cdot\rangle$ denotes the Euclidean inner product, it inherently possesses rotation and translation invariance. Consequently, $\langle\cdot, \cdot\rangle_{\mathcal{M}_p}$ inherits these desirable geometric properties. Moreover, this inner product preserves the separability of parameters $P$ and $r$, while simultaneously coupling them with $\tau$ through Equation (8). Let $\theta = (P, r, \tau)$ and $\delta\theta = (\delta P, \delta r, \delta\tau)$. Denote the Jacobian of $\beta$ by $D\beta(\theta) = \frac{\partial\beta}{\partial\theta}$. Then, the pullback metric can be finally written as follows:

$$g = \langle\delta\theta, \tilde{\delta}\theta\rangle_\Theta = \langle\delta\beta, \tilde{\delta}\beta\rangle_{\mathcal{M}_p} = \langle D\beta(\theta)\delta\theta, D\beta(\theta)\tilde{\delta}\theta\rangle_{\mathcal{M}_p} \quad (10)$$

Similarly, we can define the entire BBSC functional space $\mathcal{M}$ as follows.

**Definition 2**. $\mathcal{M}$ is called a BBSC functional space if it contains all existing $\mathcal{M}_p$, as denoted as $\mathcal{M} = \bigcup_{p\geq 0} \mathcal{M}_p$.

**Definition 3**. A mapping $\varphi : \mathcal{M}_p \to \mathcal{S}$ is deemed SE(3)-equivariant on $\mathcal{M}_p$, if $\forall\beta(P, r, \tau) \in \mathcal{M}_p$ and $\forall$ transformations $g \in$ SE(3), the following equation holds:

$$\varphi(\beta(\rho_{\mathcal{M}_p}(g)P, r, \tau)) = \rho_{\mathcal{S}}(g)\varphi(\beta(P, r, \tau)). \quad (11)$$

If $\varphi$ is infinitely differentiable with respect to $P$, $r$, and $\tau$, $\varphi$ is a $C^\infty$ mapping on $\mathcal{M}_p$, meaning it is smooth as well.

**Definition 4**. $\mathcal{GM}$ is a manifold with topological relations constructed from the direct sum of disjoint BBSC manifolds $\beta_1, \beta_2, \ldots \beta_k \in \mathcal{M}$, which is defined as follows:

$$\mathcal{GM} = \left(\bigsqcup_{i\in I} \beta_i\right) / \sim, \ \sim \subseteq \bigsqcup_{i\in I}\beta_i \times \bigsqcup_{i\in I}\beta_i. \quad (12)$$

Specifically, $\mathcal{GM}$ is a topological manifold composed of multiple $\mathcal{M}$ with certain topological relationships between them. $\mathcal{GM}$ can be viewed as a manifold with a gluing structure [42, 43, 44].

### 4.2 BBSC Manifold Construction

We follow [45] and adopt a similar BBSC fitting algorithm. Given a set of points $V \in \mathbb{R}^{N\times 3}$ from tubular objects, we apply DiffusionNet [46] (details in Appendix B.1) to embed them and output a BBSC, represented by control points $P = \{P_1, P_2, \ldots, P_n\}$, control radii $r = \{r_1, r_2, \ldots, r_n\}$, and a knot vector $\tau = \{t_1, t_2, \ldots, t_n\}$. To ensure $t_{p+2} < t_{p+3} < \cdots < t_{n-1}$ and $t_1 = \cdots = t_{p+1} = 0$, $t_n = \cdots = t_{n+p+1} = 1$, we predict the differences of the inner knot vector $\nabla\tau$ using a softmax function, and then obtain the full knot vector by applying a cumulative sum and concatenating the boundary zeros and ones. Considering all possible control parameter sets $P \in \mathbb{R}^{3\times n}$ and $r \in \mathbb{R}^n$, together with constrained knot vectors $\tau$, we construct the functional space $\mathcal{M}_p$ endowed with a smooth mapping $\beta \in C^\infty(\Theta, \mathcal{M}_p)$.

We introduce an additional template spline $\overline{\beta}$ to constrain the shape of the BBSC centerline. We minimize a loss function composed of two terms, the Hausdorff distance $D_H$ [47] between sampled

points on the constructed BBSC $\beta$ and the raw surface points, and the metric between $\beta$ and $\overline{\beta}$. Note that the Hausdorff distance can be replaced by the Chamfer distance [48], and a regularization term $\int_0^1 \kappa^2(t)\,dt$, which can be used to enforce the global smoothness of the BBSC centerline and can also be incorporated into the loss function. Based on these, we further incorporate topological relationships among the $\beta$ to construct the topological manifold $\mathcal{GM}$.

## 4.3   SE(3)-BBSCformer and Graph Convolutional Networks

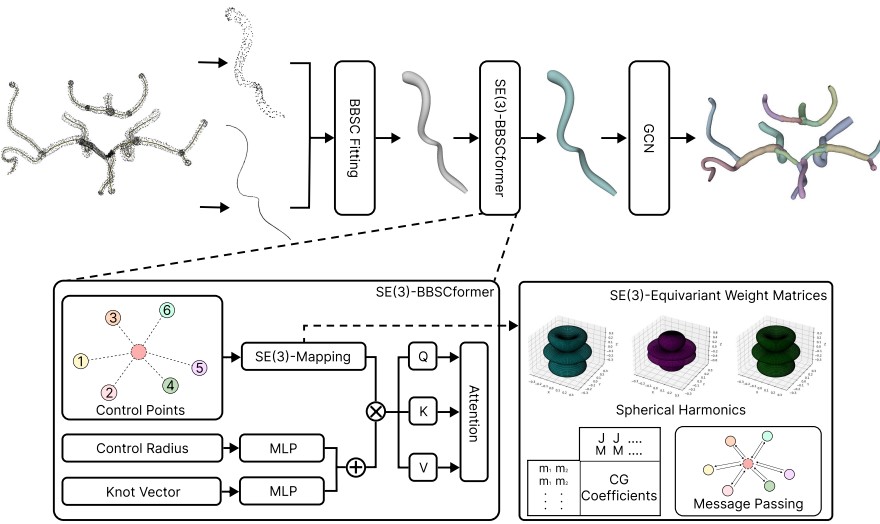

Figure 2: The SE(3)-BBSCformerGCN takes vascular centerlines and surface point clouds as inputs. A fitting module is first employed to construct BBSC. These spline-based manifold representations are then mapped into a high-dimensional manifold space via the SE(3)-BBSCformer. Finally, a GCN is utilized to process the topological relationships among the manifolds.

SE(3)-Transformer can be viewed as a mapping operating in a discrete space represented by a point cloud. In contrast, the BBSC model is a smooth and continuous representation. To integrate the advantages of both, we reconstruct the model and term it SE(3)-BBSCformer, which can be interpreted as an SE(3) mapping defined on $\mathcal{M}$. First, we compute the centroid of the control points $\overline{P} = \frac{\sum_{i=1}^n P_i}{n}$ to serve as the virtual anchoring point, acting as a central hub for transmitting information between the control points and generating the unit vector $\frac{P_i - \overline{P}}{\|P_i - \overline{P}\|}$ needed by SE(3)-BBSCformer. We initialize the $c_{i_0}$ channel feature $f_{in}^{c_{i_0}}$ as a combination of vectors generated from $r$ and $\tau$ as follows:

$$f_{in}^{c_{i_0}} = (MLP(r),\ MLP(\tilde{\tau})),\ \tilde{\tau} = ((t_1, \ldots, t_{p+1}), \ldots, (t_n, \ldots t_{n+p+1}) \in \mathbb{R}^{n \times (p+1)}.$$

Then we view each control point $P_i$ as being connected with $\overline{P}$ and replace the $x_i$ and $x_j$ with $P_i$ and $\overline{P}$ in Equation (4) and (5). The resulting SE(3)-BBSCformer is defined as follows:

$$f_{out,i}^{c_o} = W_V^{c_o c_o} f_{in,i}^{c_o} + \sum_k \alpha_{i,j} W_V^{c_o c_i} (P_i - \overline{P}) f_{in,j}^{c_i}, \tag{13}$$

where the attention matrix is computed as follows:

$$\vec{q}_i = \bigoplus_{c_o} \sum_{c_i} W_Q^{c_o c_i} f_{in,i}^{c_i},\ \vec{k}_j = \bigoplus_{c_o} \sum_{c_i} W_K^{c_o c_i} (P_j - \overline{P}) f_{in,j}^{c_i},\ \alpha_{ij} = \frac{\exp(\vec{q}_i^T \vec{k}_j)}{\sum_{j=1}^n \exp(\vec{q}_i^T \vec{k}_j)}. \tag{14}$$

For convenience, we abbreviate SE(3)-BBSCformer as the mapping $\varphi : \mathcal{M}_p \to \mathcal{S}$, where $\mathcal{S}$ denotes the high-dimensional functional space. Variable $\varphi(\beta)$ can be interpreted as the high-dimensional representation of BBSC in $\mathcal{S}$. If an object is a manifold with topological relationships, composed of multiple $\beta$ components as described in **Definition 4**, then GCN can be applied to capture the topological relationships. Each BBSC manifold $\beta \in \mathcal{GM}$ is treated as a node in the graph $G$ and

$\varphi(\beta)$ serves as the initial node feature $v$. For simplicity, we initialize the edge attribute $e$ as the difference between features of the connected node features. Then, the GCN is formulated as follows:

$$e_{i,j} = MLP([v_i, v_j, e_{i,j}]), \ v_i = MLP\left(\left[v_i, \frac{1}{\|j\|}\sum_{j \in \mathcal{N}_j} MLP([v_j, e_{i,j}])\right]\right). \quad (15)$$

Taking the CoW as an input example, we propose the SE(3)-BBSCformerGCN. The pipeline of our model is shown in Figure 2.

### 4.4 Theoretical Analysis and Discussion of Advantages

The BBSC mainfold representation $\beta$ is highly compact, allowing complex tubular shapes to be described using only a small number of control parameters and knot vector, thereby substantially reducing storage and computational costs. Moreover, the BBSC manifold exhibits strong control-lability because its shape can be easily adjusted by modifying the control parameters and the knot vector. Due to the smoothness of $\beta$, geometric and differential quantities such as normals, curvature, and torsion can be accurately calculated at any position in the object, without relying on numerical approximations that are prone to errors when estimated from discrete data [49, 50]. This property endows the representation with stronger expressive power and improved theoretical interpretability. Moreover, $\beta$ can be shown to admit continuous higher-order partial and mixed derivatives for all parameters, i.e., $\beta \in C^\infty$. This ensures that $\mathcal{M}$ forms a continuous and infinitely differentiable functional space. Benefiting from these properties, the BBSC manifold representation is particularly well-suited for modeling complex geometric tubular structures.

The SE(3)-equivariant mapping $\varphi : \mathcal{M} \to \mathcal{S}$ can be shown to be a $C^\infty$ mapping. Spherical harmonics are also $C^\infty$ [51], so the differentiability of $\varphi$ primarily depends on that of the MLP. In turn, the MLP's differentiability is determined by the choice of activation function. In this work, all MLPs are constructed using GELU activations and linear layers, both of which are $C^\infty$ mappings. Consequently, $\varphi$ is a $C^\infty$ mapping from $\mathcal{M}$ to $\mathcal{S}$ that satisfies SE(3) equivariance. The group-equivariant $\varphi$ preserves the intrinsic symmetries of $\mathcal{M}$ when mapping to $\mathcal{S}$, ensuring the stability of its shape features. Moreover, the GCN designed to capture topological relationships maintains the overall topological stability of the manifold space, thereby enhancing the model's generalization and robustness while remaining lightweight.

## 5 Experiments and Analysis

We evaluate the performance of our proposed model on the branch classification task for the CoW, a critical arterial structure located at the base of the human brain. Experiments are conducted on two datasets: the TopCoW dataset from the MICCAI 2024 Challenge [52], which contains 125 samples across 13 anatomical classes, and a clinical dataset collected from a collaborating medical institution, comprising 1,182 samples across 22 classes. We focus on the CoW branch classification task because it is a complex tubular structure characterized by both geometric and topological properties, and necessitate efficient computational approaches. Moreover, CoW exhibits a highly bilateral symmetry while simultaneously presenting complex topological heterogeneity, making group equivariant networks and GCNs particularly well-suited for modeling such structures. (see detailed descriptions of the CoW datasets, preprocess and training protocols in D)

### 5.1 Experiments on TopCoW 2024 MICCAI Challenge

We evaluate our model on the publicly available TopCoW dataset [52]. We include several SOTA voxel-based and point cloud-based classification methods as baselines. In addition, we conduct ablation studies to investigate the effects of incorporating SE(3)-equivariant BBSC mapping and the use of GCN for modeling topological relationships. Given the small sample size of the TopCoW 2024 dataset, we employ 5-fold cross-validation to enhance training robustness. We calculate the mean and standard deviation, highlighting the highest mean and lowest standard deviation in bold. Our model achieves the highest mean and lowest variance across AUC-ROC, Recall, and F1 score, while also demonstrating competitive performance in Accuracy and Precision (see results in Table 1).

We illustrates the class-wise prediction accuracy on the CoW dataset (see Figure 3). Across the majority of labels, our model consistently outperforms baseline methods. Due to the highly heterogeneous

Table 1: Performance comparison of different methods on the TopCoW 2024 dataset

| Method | Accuracy% | AUC-ROC% | Precision% | Recall% | F1 Score% |
|---|---|---|---|---|---|
| 3D DenseNet [53] | $97.02 \pm 0.57$ | $99.63 \pm 1.94$ | $95.86 \pm 0.51$ | $95.42 \pm 0.41$ | $97.47 \pm 0.48$ |
| 3D ResNet [54] | $96.54 \pm 0.54$ | $99.61 \pm 0.11$ | $96.79 \pm 0.76$ | $96.44 \pm 0.83$ | $96.52 \pm 0.80$ |
| PointNet [24] | $94.10 \pm 0.88$ | $99.46 \pm 0.20$ | $94.81 \pm 1.36$ | $94.69 \pm 0.92$ | $94.57 \pm 1.10$ |
| PointNet++ [55] | $96.61 \pm \mathbf{0.53}$ | $99.61 \pm 0.09$ | $96.72 \pm \mathbf{0.49}$ | $96.52 \pm 0.55$ | $96.45 \pm 0.60$ |
| CurveNet [56] | $97.14 \pm 1.16$ | $99.72 \pm 0.20$ | $97.31 \pm 1.16$ | $97.07 \pm 1.19$ | $97.09 \pm 1.22$ |
| RepSurf-U [57] | $97.41 \pm 0.68$ | $99.86 \pm 0.07$ | $96.83 \pm 0.52$ | $96.61 \pm 0.54$ | $96.50 \pm 0.55$ |
| PointMLP [58] | $94.10 \pm 2.27$ | $99.03 \pm 0.56$ | $94.41 \pm 2.35$ | $94.05 \pm 2.31$ | $93.94 \pm 2.40$ |
| SE(3)-Transformer [30] | $61.35 \pm 5.80$ | $90.93 \pm 1.31$ | $62.36 \pm 5.28$ | $60.77 \pm 5.70$ | $60.58 \pm 5.69$ |
| SE(3)-BBSCformer | $87.91 \pm 1.35$ | $98.24 \pm 0.31$ | $88.97 \pm 1.66$ | $88.01 \pm 1.32$ | $87.98 \pm 1.26$ |
| BBSCformerGCN | $\mathbf{97.95} \pm 1.22$ | $99.89 \pm 0.02$ | $97.25 \pm 0.79$ | $98.34 \pm 0.60$ | $97.78 \pm 0.70$ |
| SE(3)-BBSCformerGCN | $97.53 \pm 0.83$ | $\mathbf{99.99} \pm \mathbf{0.01}$ | $\mathbf{97.36} \pm 0.53$ | $\mathbf{98.44} \pm \mathbf{0.31}$ | $\mathbf{97.87} \pm \mathbf{0.41}$ |

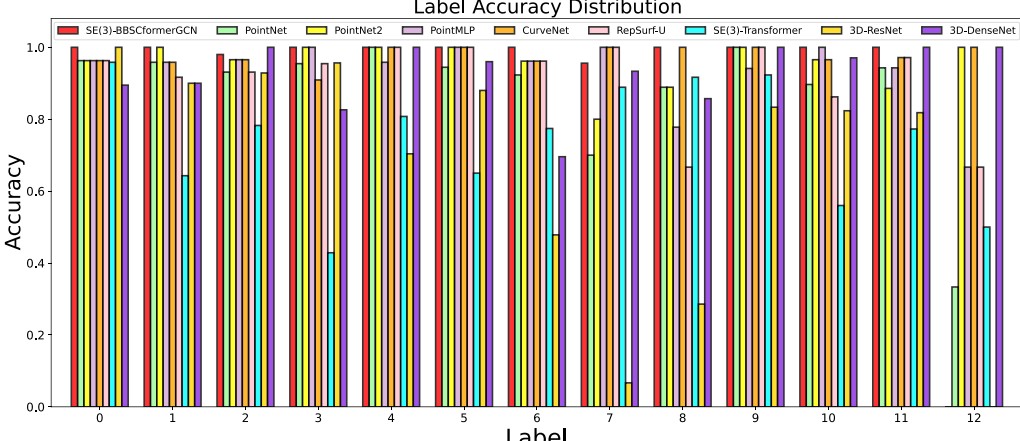

Figure 3: Per-class prediction accuracy of various models evaluated on TopCoW 2024 dataset.

anatomical structure of the CoW [59, 60, 61], the distribution of different branches label is notably imbalanced. For example, in the TopCoW 2024dataset, only 5% of the 125 samples contain the segment corresponding to Label 12 (see details in Appendix D.1). This extreme scarcity introduces a high degree of randomness during testing. However, we observe that this issue is effectively mitigated by increasing the total number of training samples, as further demonstrated in Section 5.2.

## 5.2 Experiments on Real-World Clinical Data

Table 2: Performance comparison of different methods on the clinical dataset

| Method | Accuracy% | AUC-ROC% | Precision% | Recall% | F1 Score% |
|---|---|---|---|---|---|
| PointNet [24] | $69.20 \pm 0.98$ | $96.50 \pm 0.24$ | $70.02 \pm 0.77$ | $69.17 \pm 0.96$ | $68.90 \pm 1.02$ |
| PointNet++ [55] | $70.27 \pm 0.16$ | $96.60 \pm 0.17$ | $70.82 \pm 0.43$ | $70.28 \pm 0.20$ | $70.05 \pm \mathbf{0.14}$ |
| CurveNet [56] | $81.91 \pm 0.14$ | $93.80 \pm 1.13$ | $62.92 \pm 3.14$ | $60.13 \pm 4.25$ | $58.51 \pm 4.58$ |
| RepSurf-U [57] | $78.22 \pm 0.17$ | $98.40 \pm 0.11$ | $78.34 \pm 0.20$ | $77.99 \pm 0.29$ | $78.00 \pm 0.25$ |
| PointMLP [58] | $82.70 \pm \mathbf{0.10}$ | $98.66 \pm \mathbf{0.06}$ | $82.86 \pm \mathbf{0.14}$ | $82.70 \pm \mathbf{0.10}$ | $82.67 \pm \mathbf{0.14}$ |
| SE(3)-Transformer [30] | $61.96 \pm 1.45$ | $92.32 \pm 0.96$ | $61.05 \pm 2.07$ | $61.94 \pm 1.47$ | $60.77 \pm 2.57$ |
| SE(3)-BBSCformer | $71.06 \pm 1.39$ | $93.25 \pm 0.60$ | $70.25 \pm 1.16$ | $71.04 \pm 1.39$ | $70.45 \pm 1.21$ |
| BBSCformerGCN | $95.45 \pm 1.09$ | $99.34 \pm 0.21$ | $95.10 \pm 1.48$ | $95.38 \pm 1.13$ | $95.23 \pm 1.29$ |
| SE(3)-BBSCformerGCN | $\mathbf{96.11} \pm 1.01$ | $\mathbf{99.38} \pm 0.19$ | $\mathbf{96.11} \pm 0.93$ | $\mathbf{96.08} \pm 0.97$ | $\mathbf{96.02} \pm 1.05$ |

We further validate the generalization ability of our model on clinical data collected from the collaborating medical institution. This dataset contains approximately ten times more number of samples than the TopCoW 2024 MICCAI Challenge dataset. Although the inherent heterogeneity

of CoW still leads to imbalanced class distributions, the substantially larger sample size helps alleviate the randomness in predictions for underrepresented classes. The results presented in Table 2 demonstrate that our model exhibits stronger generalization ability compared to voxel-based and point-based approaches. While other methods experience a performance drop of 10–20%, our model maintains a consistently high level of evaluation metrics. Notably, SE(3)-BBSCformerGCN outperforms BBSCformerGCN in both mean and standard deviation across all evaluation metrics, further validating the stability and generalization advantages introduced by the SE(3)-equivariant mapping. Additionally, the comparison between SE(3)-BBSCformerGCN and SE(3)-BBSCformer highlights the critical role of topological information. Collectively, these findings suggest that both geometric and topological stability are the key factors underpinning the generalization capability of our model. Similarly, we provide the per-class prediction accuracy across all label categories (Figure 4). As shown, our model consistently outperforms all baselines across all 22 classes.

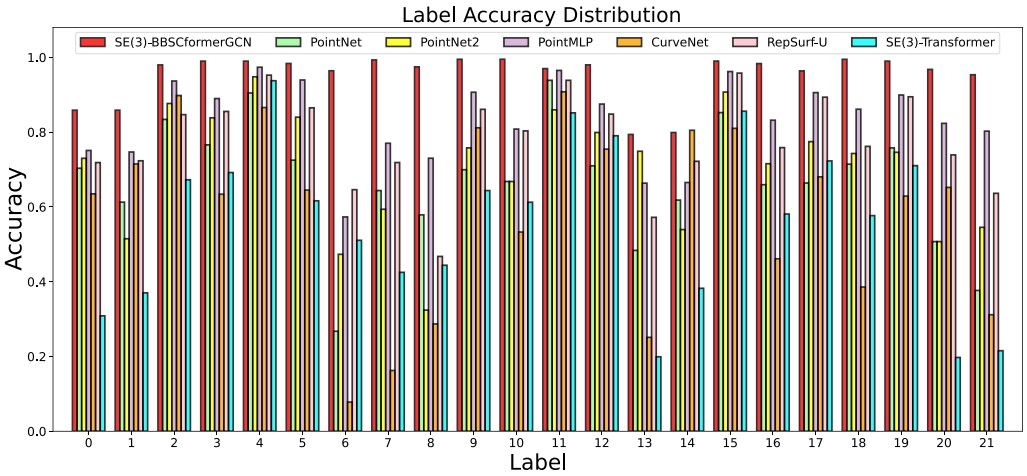

Figure 4: Per-class prediction accuracy of various models evaluated on real-world clinical dataset.

## 5.3    Training Efficiency, Stability, and Computational Cost Analysis

We illustrates the training and testing accuracy curves on the clinical datasets (see Figure 5). Our method consistently achieves faster convergence and greater stability across all methods. These advantages are largely attributed to the compact input representation, which preserves fine-grained geometric details without the need for aggressive downsampling. More critically, while other models exhibit severe overfitting, both SE(3)-BBSCformer and the SE(3)-Transformer demonstrate strong resistance to overfitting, highlighting the robustness and generalization power of equivariant architectures. Moreover, leveraging BBSCs to capture complex geometric features and topological structures of tubular systems further enhances our model's performance over existing baselines.

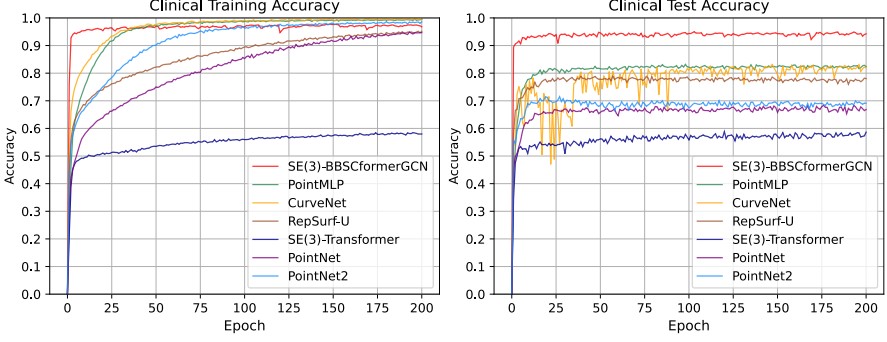

Figure 5: Epoch-wise training and testing accuracy evaluated on the clinical dataset.

To validate the efficiency of BBSC for tubular object representation, we compared our model against several baselines on evaluation time, FLOPs, and parameter count (see Table 3). SE(3)-

BBSCformerGCN achieves superior generalization and classification performance while significantly reducing inference time and memory usage. These results underscore the practical benefits of our design, particularly for fast and cost-effective deployment in clinical settings.

Table 3: Parameters amount and computing cost comparison of different methods

| Method | Evaluation Time(ms) | FLOPs(M) | Parameters(M) |
|---|---|---|---|
| 3D DenseNet [53] | 8.03 | 9743.76 | 18.56 |
| 3D ResNet [54] | 10.95 | 11463.08 | 85.23 |
| PointNet [24] | 0.88 | 450.38 | 3.46 |
| PointNet++ [55] | 1.54 | 4067.53 | 1.74 |
| CurveNet [56] | 81.95 | 269.56 | 2.13 |
| RepSurf-U [57] | 59.84 | 911.32 | 1.48 |
| PointMLP [58] | 38.39 | 15733.95 | 13.23 |
| SE(3)-Transformer [30] | 19.03 | 456.33 | 0.12 |
| SE(3)-BBSCformer | 1.92 | 1.72 | 0.35 |
| BBSCformerGCN | 1.14 | 88.98 | 3.60 |
| SE(3)-BBSCformerGCN | 3.02 | 59.01 | 2.70 |

## 6   Discussion and Future Work

We leverage an extension of B-spline curves to the spherical domain, which constructs the BBSC, to model tubular structures. We formulate a mathematically well-defined smooth manifold based on this representation, construct a functional space for tubular manifold, and discuss the associated metric on this space. Furthermore, we propose a rotation and translation equivariant mapping, which is applied to the anatomical classification of the CoW. While our experiments focus on this clinically significant cerebrovascular structure, the proposed framework is readily applicable to a wide range of other tubular structures. Additionally, the BBSC structure is naturally extensible. The control vectors can be lifted into an n-dimensional space, where the B-spline basis functions can be reused to construct smooth and high-dimensional manifolds. Notably, SplineCNN [62] has previously utilized this insight to define smooth convolutional filters via B-Spline kernels. Lastly, our model maps $\mathcal{M}$ defined by BBSC into a high-dimensional space $\mathcal{S}$, preserving SE(3)-equivariance of the 3D parameters $P$. However, a limitation of our current model is that the graph convolution operations used to model topological relations on $\mathcal{S}$ are not equivariant, which could compromise the overall stability and full equivariance of the model. It is worth investigating into the design of equivariant mappings in high-dimensional topological manifold spaces as a line of future work.

## 7   Conclusion

We propose SE(3)-BBSCformerGCN, a novel deep learning architecture that achieves group equivariance on tubular manifolds. Our model takes as input a BBSC manifold, parameterized by a small set of 4D control parameters and knot vectors. This representation not only enables more effective extraction of complex geometric and topological features from tubular structures, but also reduces computational complexity. This design offers strong mathematical interpretability and improved computational efficiency. We validate our model on a clinically important yet computationally underexplored task: classifying anatomical configurations of the CoW, and demonstrate its superior performance in classification, generalization, and resistance to overfitting. Our code will be available on https://github.com/niuyixuan/SE-3–BBSCformerGCN.

## 8   Acknowledgments

This research is partially supported by the National Natural Science Foundation of China (Grant No. 62072045), the Beijing Municipal Science and Technology Commission and the Zhongguancun Science Park Management Committee (Grant No. Z221100002722020), the Natural Science Foundation of Beijing (Grant No. 7242167), and the Teaching Reform Project of Beijing Normal University. We would also like to express our sincere gratitude to Peking Union Medical College Hospital (PUMCH) for kindly providing the clinical CoW scan data used in this study.

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

# A  Background

## A.1  Geometric Deep Learning and Continuous Function Learning

Geometric deep learning [27] is a branch of deep learning that leverages deep neural networks to model and extract intrinsic geometric features of structured data. To date, geometric deep learning has yielded impressive results in a range of fields, notably physics, biology, and computer graphics. In physiscs, GNS [63] and MESHGRAPHNET [64] investigate the application of geometric deep learning in the simulation of fluid and cloth, aiming to capture their intricate physical dynamics. In biology, DeepDock [65] and its enhanced version, caDeepDock [66], predict the binding conformation by learning the distributions based on Euclidean distance. In computer graphics, PointNet [24] and PointNet++ [55] represent major breakthroughs in point cloud processing by effectively capturing geometric features directly from raw point clouds. Most geometric deep learning methods operate on discrete data in Euclidean space, such as point clouds, by constructing geometric graphs[67, 68] or defining local point neighborhoods. These approaches typically assign geometric attributes (e.g., spatial coordinates) to the nodes of the graph and the points, enabling the extraction of meaningful geometric features.

A topological manifold [69] is a special type of geometric space, formally defined as a topological space that is locally Euclidean. Compared to modeling discrete data such as point cloud in 3D Euclidean space, modeling directly on continuous manifold domains can avoid the errors introduced by numerical estimation or implicit representations learned via neural networks. This approach enables models to achieve greater expressiveness, interpretability, and generalization capability. Function learning models [70, 71], with the goal of constructing high-dimensional mappings from functions to functions, preserve the analytical properties of the functions themselves, such as smoothness, and has found increasing application [70, 71] . Inspired by the aforementioned approaches, we construct a 3D rotation and translation equivariant mapping on smooth surface manifolds represented by Ball B-Spline Curve(BBSC) [8]. Owing to its compactness, local modifiability, smoothness, convex hull property, the BBSC provides an efficient and accurate representation of tubular geometric structures.

## A.2  Circle of Willis

CoW is a ring-shaped arterial structure located at the base of the brain in birds, reptiles, and mammals [72]. It supplies blood to the brain and surrounding structures by connecting the bilateral internal carotid artery (ICA) systems with the vertebrobasilar system. In cases of stenosis or occlusion of a major artery (e.g., the ICA), the CoW provides collateral circulation to maintain perfusion in ischemic regions, serving a compensatory function. Additionally, the CoW plays a critical role in balancing and regulating intracranial arterial pressure and in protecting against acute cerebrovascular events [73]. Notably, the CoW is also a frequent site for cerebral aneurysms, which, when ruptured, can lead to stroke and potentially fatal outcomes. In its idealized form, the standard CoW consists of several key arteries: the anterior cerebral arteries (left and right), the anterior communicating artery, the distal termini of the internal carotid arteries (left and right), the P1 segments of the posterior cerebral arteries (left and right), and the posterior communicating arteries (left and right) [74]. Depending on clinical needs, more detailed sub-regional definitions may be adopted. However, due to factors such as genetic variation and developmental anomalies, the CoW often exhibits significant topological heterogeneity in its anatomical structure. For example, incomplete CoWs may result from the absence of one or more arterial segments; developmental variants may manifest as arteries that are duplicated, abnormally narrow, or overly dilated. From a topological perspective, the CoW is generally classified into three categories: complete circles, partially closed, and open configurations. Among them, open CoWs are often associated with poor collateral capacity and are more prone to widespread ischemia or infarction in the presence of arterial narrowing, occlusion, or aneurysm rupture. This high degree of topological variability poses considerable challenges for clinical diagnosis and increases the importance of automated and robust computational methods.

The segmentation and classification of the CoW are both critical and well-suited to computational approaches. Accurate and efficient methods not only reduce the workload of medical professionals but also enhance diagnostic precision, playing an essential role in the detection and early monitoring of CoW-related cerebrovascular diseases. To the best of our knowledge, there are currently few computational models specifically designed for CoW segmentation and classification. Although some methods developed for general vascular analysis can be adapted, their performance on CoW remains

unstable due to its inherent topological heterogeneity [59, 60, 61]. Therefore, designing effective models for CoW-related tasks requires a joint analysis of both geometric and topological features. Such models [75, 9, 76] are crucial for capturing the structural variations and delivering reliable diagnostic support in real-world clinical scenarios.

# B  Preliminary Model Architecture

## B.1  DiffusionNet

DiffusionNet [46] is an effective method to learn point cloud features using the principle of heat diffusion. It mainly consists of two modules: diffusion range learning and gradient feature learning.

**Learning diffusion**. The diffusion learning can be interpreted as a filter that allows each point to propagate information radially and the diffusion range $H_t(u_0)$ with initial state $u_0$, mainly depends on a learnable diffusion time $t$, is described by formula $H_t(u_0) = e^{t\Delta}u_0$. DiffusionNet effectively discretizes and simulates the diffusion process using the implicit Euler time step method,

$$h_t(u) = (M + tL)^{-1}Mu \tag{16}$$

where $M$ and $L$ represent mass matrix and weak Laplace matrix. To avoid the inconvenience of solving a large linear system separately for each channel,DiffusionNet offers spectral acceleration which leveages the precomputed basis of low frequency Laplacian eigenfunctions to represent the diffusion approximately. Let $\Phi = [\phi_i] \in \mathbb{R}^{V \times k}$ represent the concatenated matrix of the eigenvectors corresponding to the first k smallest eigenvalues $\lambda_i$ where $[\phi_i]$ are solutions to $L\phi_i = \lambda_i M\phi_i$. The coefficients $c$ of spectral basis can be obtained by $c = \Phi^T Mu$ and $u$ can be recovered by $u = \Phi c$, and the coefficients $c_i^t$ after $t$ time diffusion can be calculated by formula $c_i^t = e^{-\lambda_i}c_i^0$. Then diffusion can be simplified as follows,

$$h_t(u) = \Phi c^t = \Phi(e^{-\lambda_0 t}, e^{-\lambda_1 t}, \ldots, e^{-\lambda_k t})^T \cdot (\Phi^T Mu) \tag{17}$$

**Learning gradient feature**. In additional, DiffusionNet constructs a learning scaling gradient feature for each point to enhance the feature learning ability of the filter. First, the gradient operator which is assembled into a sparse matrix $G$. The matrix is independent of the vertex feature and can be precomputed for each shape by computing the least-squares approximation of the function values of each point and its neighbors on the tangent plane. Note that in order to conveniently represent the tangent vector in the calculation of the gradient matrix, DiffusionNet uses a complex expression. Multiply each channel feature $u$ of vertex $v$ by the gradient matrix $G$ and stack them up to get the gradient feature $w_v = [Gu]$. Then a learned scaling gradient feature $g_v$ shown as follow:

$$g_v = tanh(Real(\langle \overline{w}_v, Aw_v \rangle)) = tanh\left(Real\left(\sum_{i=1}^{D}\sum_{j=1}^{D}\overline{w}_v(i)A_{i,j}w_v(j)\right)\right), \ A \in \mathbb{R}^{D \times D} \tag{18}$$

where $A$ is the learned real matrix which describe the scaling and $D$ is the number of the feature channel. Whether to choose a real or complex matrix A mainly depends on whether the point has a normal. Because multiplying a complex scalar do both the scaling and rotationtransformations. The real part of the scalar represents scaling, while the imaginary part represents rotation. Generally, the direction of rotation deponds on the outward normal and orientation, so for points without normal, a real matrix $A$ is sufficient.

It is worth noting that even though only low-frequency basis is used in the diffusion spectral acceleration part, DiffusionNet can still learn high-frequency features through gradient learning and MLPs (Multi-Layer Perceptrons).

## B.2  SE(3)-Transformer

The SE(3)-Transformer [30] integrates the principles of 3D rotation and translation equivariance by embedding Tensor Field Network (TFN) [29] into the computation of the key, query, and value vectors within the Transformer architecture. This design endows the Transformer with SE(3) equivariance in the 3D Euclidean space, ensuring that its outputs are consistent under rigid transformations. First, the SE(3)-Transformer leverages spherical harmonics [37] to map 3D point coordinates into a higher-dimensional representation space. Spherical harmonics form a set of orthogonal Fourier bases defined

on the unit sphere $S^2$. Specifically, an $l$ degree spherical harmonics $Y^l$ can map any 3D unit vector to a $2l+1$-dimensional vector in $\mathbb{R}^{2l+1}$, expressed as $Y^l(\vec{x}) = \{Y^l_{-l}(\vec{x}), Y^l_{-l+1}(\vec{x}), \ldots, Y^l_{l-1}(\vec{x}), Y^l_l(\vec{x})\}$. Then, the Wigner-D matrices [77], which are unitary and irreducible, are introduced to achieve equivariance of high-dimensional vectors generated by spherical harmonics under rotations in SO(3):

$$Y^l(R_g\vec{x}) = W^l(g)Y^l(\vec{x}), \quad g \in SO(3), R \in \mathbb{R}^{3\times3}, W^l(g) \in \mathbb{R}^{(2l+1)\times(2l+1)}. \quad (19)$$

Furthermore, Clebsch-Gordan tensor product [38] provides a mechanism to aggregate vectors from spaces of different dimensions while preserving equivariance. For $\vec{V}^{l_1} \in \mathbb{R}^{(2l_1+1)\times C_1}$ and $\vec{V}^{l_2} \in \mathbb{R}^{(2l_2+1)\times C_2}$, $\vec{V}^l = \vec{V}^{l_1} \otimes^l_{cg} \vec{V}^{l_2}$ can be expanded as:

$$V^l_{m,c} = \sum_{c_1=1,c_2=1}^{C_1,C_2} w_{c_1,c_2,c} \sum_{m_1=-l_1}^{l_1} \sum_{m_2=-l_2}^{l_2} Q^{(l,m)}_{(l_1,m_1)(l_2,m_2)} v^{(l_1,c_1)}_{m_1} v^{(l_2,c_2)}_{m_2}, \quad (20)$$

where $Q^{(l,m)}_{(l_1,m_1)(l_2,m_2)}$ are the Clebsch-Gordan coefficients.

SE(3)-Transformer constructs a set of learnable radial functions $\psi : \mathbb{R}_{\geq 0} \to \mathbb{R}$ and leveages the Clebsch-Gordan coefficients to form the basis kernel $W^{c_o c_i} \in \mathbb{R}^{(2c_o+1)\times(2c_i+1)}$ which can transform $(2c_i + 1)$ channels input feature to $(2c_o + 1)$ channels output feature:

$$W^{c_o c_i}(\vec{x}) = \sum_{l=|c_i-c_o|}^{c_i+c_o} \psi^{c_o c_i}_l(\|\vec{x}\|) \sum_{m=-l}^{l} Y^l_m(\frac{\vec{x}}{\|\vec{x}\|})Q^{c_o c_i}_{lm}, \quad (21)$$

where $Y^l_m$ is the $m$th element in $Y^l$, $Q^{c_o c_i}_{lm} \in \mathbb{R}^{(2c_o+1)\times(2c_i+1)}$. With the basis kernel $W^{c_o c_i}$, $c_i$ channels input feature $f^{c_i}_{in}$ of points $x$, SE(3)-Transformer can generate a $c_o$ channels feature $f^{c_o}_{out}$, as shown in Equation (4) and (5).

## C  Proof of Infinite Differentiability in BBSC Mapping

**Proposition 1**. Let $U = \{P \in \mathbb{R}^{3\times n}, r \in \mathbb{R}_{>0}, \tau \in \mathbb{T}\}$, where $\mathbb{T} = \{t_i \in [0,1]|t_{p+1} < t_{p+2} < \cdots < t_n, t_1 = \cdots = t_{p+1} = 0, t_n = \cdots = t_{n+p+1} = 1\}$. Then, the mapping $\beta : U \to \mathcal{M}$, defined according to Equation (1) and Equation (2), is a $C^\infty$ mapping.

**Proof.** The p-degree BBSC manifold can be written as:

$$\beta(P,r,\tau) = \sum_{i=0}^{n} N_{i,p}(t)\,(P_i \,;\, r_i) = \left(\sum_{i=0}^{n} N_{i,p}(t)P_i \,;\, \sum_{i=0}^{n} N_{i,p}(t)r_i\right) = (\sigma(t), \sigma(t)), \quad (22)$$

and the basis function is as follows:

$$N_{i,p}(t) = \frac{t - t_i}{t_{i+p} - t_i} N_{i,p-1}(t) + \frac{t_{i+p+1} - t}{t_{i+p+1} - t_{i+1}} N_{i+1,p-1}(t), N_{i,0} = \begin{cases} 1, & \text{if } t_i \leq t < t_{i+1}, \\ 0, & \text{otherwise.} \end{cases}, \quad (23)$$

where $P_i \in \mathbb{R}^3, r_i \in R, t_{p+1} < t_{p+2} < \cdots < t_n, t_1 = \cdots = t_{p+1} = 0, t_n = \cdots = t_{n+p+1} = 1$. We prove the smoothness of the manifold $\mathcal{M}$ by showing that $\beta$ admits infinitely many continuous partial derivatives with respect to $P$, $r$, and $\tau$.

$\beta$ is a linear combination of the control points $P_i$, and thus admits infinitely many continuous partial derivatives with respect to each $P_i$, i.e., $\forall P_i \in \mathbb{R}^3$,

$$\frac{\partial\beta}{\partial P_i} = N_{i,p}(t), \quad \frac{\partial^2\beta}{\partial P_i^2} = \cdots = \frac{\partial^\infty\beta}{\partial P_i^\infty} = 0. \quad (24)$$

Similarly, since $\beta$ admits infinitely many continuous partial derivatives with respect to each control radius $r_i$.

For $\tau$, to avoid the denominator being 0, we do not consider the endpoints of the BBSC which means we only consider the differentiability of strictly increasing sequence $t_{p+1}, t_{p+2}, \ldots, t_n$. Now we need to prove that the corresponding basis function $N_{i,p}$ which contains $t_i$ has continuous infinite partial derivatives. As observed in Equation (23), the basis function $N_{i,p}$ is a recursively defined composite function, and its differentiation can be handled via the chain rule. Since each knot value $t_i$ appears in the basis function only through two types of expressions, simply denoted as $f_1 = \frac{t-t_i}{a-t_i}$

and $f_2 = \frac{t-b}{t_i-b}$ (maybe the former needs to be added with a minus sign, but this does not affect the derivative calculation), it is sufficient to verify whether both types possess infinitely many continuous partial derivatives. Thus $\beta$ admits infinitely many continuous partial derivatives with respect to knot vector $\tau$ and can be calculated as follows:

$$\frac{d^m f_1}{dt_i^m} = -\frac{m!}{(a-t_i)^{m+1}}, \quad \frac{d^m f_2}{dt_i^m} = (-1)^m \frac{m! \cdot (t-b)}{(t_i-b)^{m+1}}, \quad a \neq t_i, \quad b \neq t_i. \tag{25}$$

By the chain rule, any higher-order derivative of $N_{i,p}$ with respect to $t_i$ can be expressed as a combination of products and sums of the derivatives of the $f_1$ and $f_2$.

For $P$ and $\tau$, the function $\beta$ is separable with respect to these variables. Specifically, it can be written as the product of $G(P) = P$ and $N(\tau) = N_p(t)$:

$$\beta = N(\tau)G(P)^T. \tag{26}$$

Its mixed partial derivatives are then expressed as

$$\frac{\partial^l \beta}{\partial P^m \partial \tau^n} = \frac{\partial^m \beta}{\partial P^m} \frac{\partial^n \beta}{\partial \tau^n} = \frac{d^m G}{dP^m} \frac{d^n N}{d\tau^n}, \quad \text{where } m+n = l. \tag{27}$$

Since the partial derivatives of $G(P)$ and $N(\tau)$ exist for all orders and are continuous, the mixed partial derivatives of $\beta$ with respect to $P$ and $\tau$ are also continuous for all orders.

Similarly, arbitrary-order continuous mixed partial derivatives with respect to $r$ and $\tau$ can be obtained. For $P$ and $r$, we have

$$c(P) = N_p(t)P^T = N_p(t)P^T \cdot 1 = c(P)H(r), \quad \text{where } H(r) = 1, \tag{28}$$

which ensures that $\beta$ also admits arbitrary-order continuous mixed partial derivatives with respect to $P$ and $r$. The same reasoning applies to $r$ itself. Thus, it can be proven that $\forall \beta \in \mathcal{M}, \beta$ is a $C^\infty$ mapping.

## D   Details of the CoW Dataset and Training Procedure

### D.1   Per-Class Distribution of Dataset

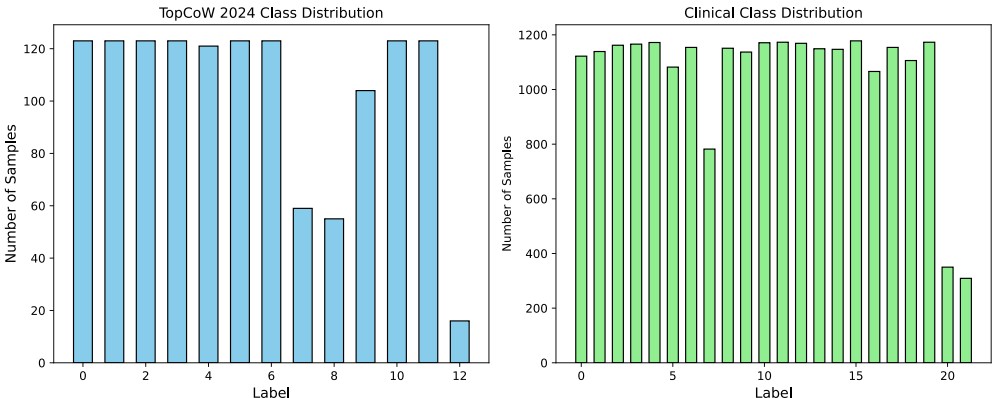

Figure 6: Per-class distribution of TopCoW 2024 and clinical dataset.

The CoW exhibits a highly heterogeneous anatomical structure, which arises not only from inter-ethnic and inter-population variability[78] but also from genetic factors[79], embryonic development[80], environmental influences[81], and hemodynamics-driven vascular remodeling[82]. This inherent heterogeneity leads to significant class imbalance in data representations of the CoW. As illustrated in Figure 6 we present an overview of the class distribution in both the TopCoW 2024 MICCAI Challenge dataset and our internal clinical dataset. Notably, in the TopCoW dataset, category 12 is severely underrepresented, with vessels labeled as class 12 comprising only 5% of all samples. Even

when training the model with a uniformly stratified split between training and testing sets, the model's learning of class-12 vessels remains highly stochastic—e.g., in extreme cases, the test set might contain only a single sample with class-12 branches. In contrast, as shown in Figure 4, our method demonstrates robust performance on the clinical dataset, even for rare classes such as 20 and 21, achieving accurate segmentation despite their limited presence in the data. While data augmentation is a common strategy to address such imbalance, we currently do not employ it. This is primarily due to the use of SE(3)-equivariant architectures, which are inherently invariant to translations and rotations, rendering many conventional augmentations redundant in our context.

## D.2 Preprocess of Clinical Data

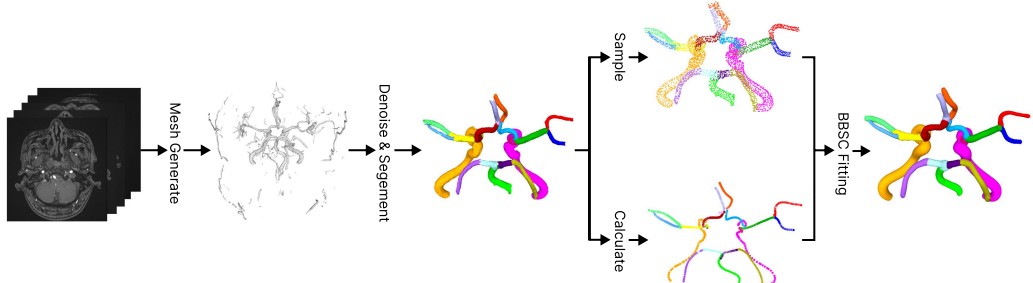

Figure 7: Clinical data preprocessing pipeline. A mesh is constructed from each MRI scan. The denoise and clean operations are performed to remove noise. Then the branches of the CoW are segmented. Surface point clouds and centerlines are computed from the segmented mesh, followed by BBSC fitting.

Compared with the TopCoW2024 dataset, clinical CoW scans are more challenging due to imaging artifacts and spurious small vessel components. Consequently, a series of preprocessing steps is required before BBSC fitting and shown in Figure 7. First, when severe MRI artifacts such as motion blur or metal-induced distortions degrade mesh quality, denoising methods can be applied [83]. Vascular meshes are then reconstructed from MRI either using geometric methods [84, 85] or the VMTK package [86]. The reconstructed meshes often contain noise from small vessel components and exhibit disconnected fragments. These are removed through cleaning [87] and repaired using mesh fix algorithms [88]. Branch segmentation [89, 90] and ground truth labeling follow, with manual verification and correction by domain experts to ensure accuracy. The majority of preprocessing operations can be implemented with open-source packages such as VMTK or medical CAD tools [91, 92]. Finally, surface point clouds and centerlines are extracted from the cleaned meshes for BBSC fitting. It is worth noting that, in our pipeline, the mesh is first reconstructed and then subjected to cleaning, denoising, and segmentation. Alternatively, these preprocessing operations can also be applied directly to the MRI images prior to mesh reconstruction.

## D.3 Training Detail

**BBSC construction.** For each branch of the CoW, we individually construct a BBSC representation. These are then processed using DiffusionNet to obtain the three key components of the BBSC: control points, control radius, and the knot vector. For notational convenience, we denote the concatenation of control points and radius as the control parameters, represented as 4D vectors. In general, the length of the control parameters is either proportional to the branch's arc length or determined by a threshold-based allocation strategy. However, in our experiments, we observed that assigning a fixed control parameter length of 13 for all branches significantly simplified and accelerated the BBSC construction process, while having negligible impact on downstream classification performance. When constructing BBSCs, we consistently set the spline degree to 3. For the TopCoW 2024 dataset, we assign all branches a control parameter length of 13. For the clinical dataset from hospital, we adopt a length allocation scheme based on arc length thresholds, assigning control parameter lengths of 13, 10, and 5, respectively. Additionally, during the BBSC fitting process, we adopt the computationally efficient and easy-to-implement L2 norm metric (Equation (7)). The term $\|P - \tilde{P}\|_2$ in this metric can also be replaced with $\langle P, \tilde{P} \rangle$, yielding a comparable fitting performance. The

weighting parameters $\alpha$ and $\eta$ are set to 1. During training, all control parameters are padded to a fixed length of 13 to ensure consistency across samples. It is important to note that we pad with zeros. To distinguish between padded zeros and the zero entries in the knot vector, we apply a value shift of +1 to all elements in the knot vector.

**SE(3)-BBSCformerGCN.** Due to the local support property of the $p$-degree B-Spline, where the influence of the $i$-th control parameter is restricted to $[t_i, \dots, t_{i+p+1}]$, we first construct an $n \times (p+1)$ matrix $\tilde{\tau}$, where the $i$-th row corresponds to $[t_i, \dots, t_{i+p+1}]$ $(p = 3)$. We then apply several MLP jointly to $\tilde{\tau}$ and the control radius to produce a $(2l + 1)$-dimensional vector, where $l = 0, 1, \dots, c_{i_0}$ $(c_{i_0} = 3)$. The direct sum of $c_{i_0}$ vectors constitutes the rotation and translation equivariant input $f_{in}^{c_{i_0}}$ of SE(3)-BBSCformer defined in Section 4.3. The degree of spherical harmonics is also $c_{i_0}$. The architecture of our graph convolutional network (GCN) follows a hybrid of the GCN and Res-GCN design used in DeepDock, as illustrated in Figure 8 (we set $H_1 = 3$ and $H_2 = 10$).

$$\longrightarrow H_1 \times \boxed{\text{GCN}} \longrightarrow H_2 \times \boxed{\textbf{Residual GCN}} \longrightarrow$$

Figure 8: Architecture of GCN in SE(3)-BBSCformerGCN.

**Train and test detail.** We train and evaluate all models presented in this paper on a single NVIDIA 3090Ti GPU. For the SE(3)-BBSCformerGCN, we set the number of SE(3)-BBSCformer layers to 1. During training, we used the Adam optimizer with a fixed learning rate of 0.001 and a weight decay of 0.01. The number or workers is 1. The loss function is set to cross-entropy for all models, and the number of training epochs is fixed at 200. For the TopCoW 2024 dataset, lightweight models such as SE(3)-BBSCformer, BBSCformerGCN, and SE(3)-BBSCformerGCN are trained and tested using a 5-fold cross-validation protocol, and their mean and standard deviation is computed and showed in Figure 3. During training, we split the TopCoW 2024 dataset into 100 samples for training and 20 for testing. Similarly, the clinical dataset is partitioned into training and test sets with a 4:1 split.

# E  Per-Class AUC-ROC

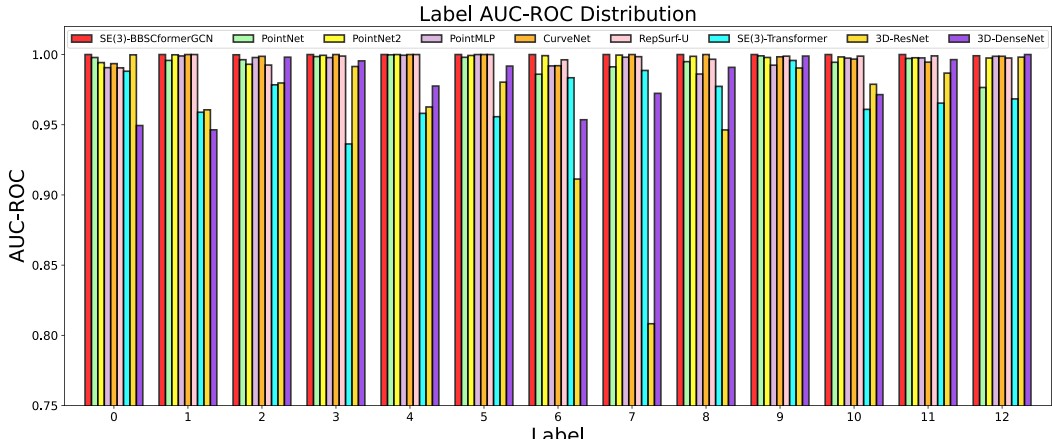

Figure 9: Per-class AUC-ROC of various models evaluated on the TopCoW 2024 Circle of Willis dataset.

We further present the AUC-ROC scores of different models across various label categories on both the TopCoW 2024 dataset and a clinical dataset, aiming to assess the per-class classification performance in more detail. As shown in Figure 10 and Figure 4, the y-axis is restricted to the range of 0.75 to 1.0 for improved visual clarity, allowing for more effective comparison between models in high-performance regimes. SE(3)-BBSCformerGCN consistently achieves the highest AUC-ROC scores across all anatomical label categories, clearly outperforming baseline methods such as voxel-based and point-based networks. This consistent superiority is observed not only on the relatively small-scale TopCoW 2024 dataset, but also on the large-scale real-world clinical dataset,

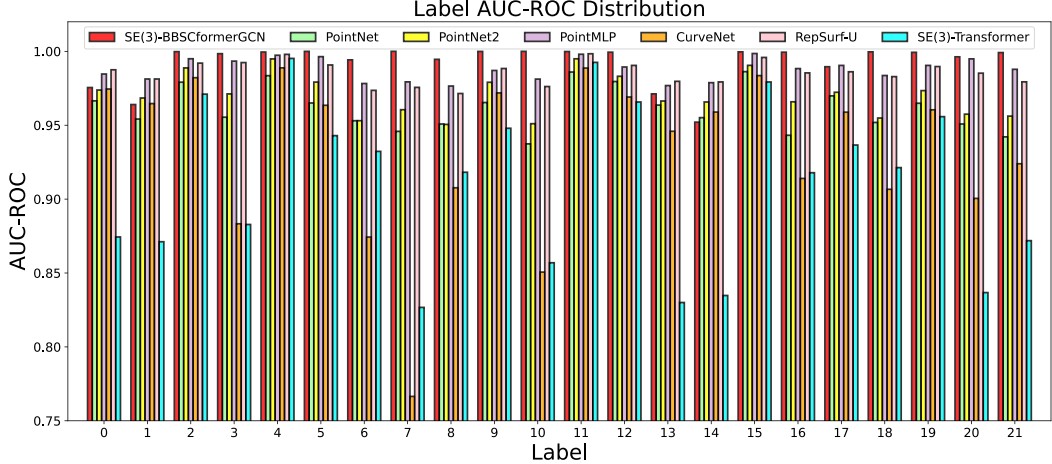

Figure 10: Per-class AUC-ROC of various models evaluated on the Circle of Willis dataset collected from clinical imaging at hospital.

highlighting the model's strong generalization ability. The results further demonstrate the advantages of our BBSC representation and the SE(3)-equivariant architecture, which together enable precise modeling of complex geometric and topological structures while maintaining robustness across domains. Importantly, even for challenging categories with high anatomical variability, our model maintains high AUC-ROC scores, indicating its effectiveness in capturing fine-grained structural differences and mitigating overfitting.

## F    Case Study of Clinical Data

We conduct a case study on Circle of Willis (CoW) clinical data collected from a collaborating medical institution, focusing on samples where SE(3)-BBSCformerGCN makes prediction errors and which exhibit certain topological heterogeneity. As shown in Figure 11, the first row presents a complete CoW structure, a configuration that only accounts for about 20%–25% of real-world cases. In the remaining samples with topological variations, most commonly observed is the absence of the bilateral posterior communicating arteries (PcoA-L and PcoA-R), which connect the upper and lower parts of the CoW. In the fourth row, a more severe case of unilateral absence is observed.

Beyond topology, geometric heterogeneity within the same type of branch also poses challenges. For instance, the MCA-L2 branch highlighted in green in the third row is noticeably longer than its counterparts in other samples, while the ICA-R1 branch in the fourth row, marked in ochre, is significantly thicker than in the other cases. Such variations in both topology and geometry contribute to the clinical difficulty of accurately classifying CoW branches.

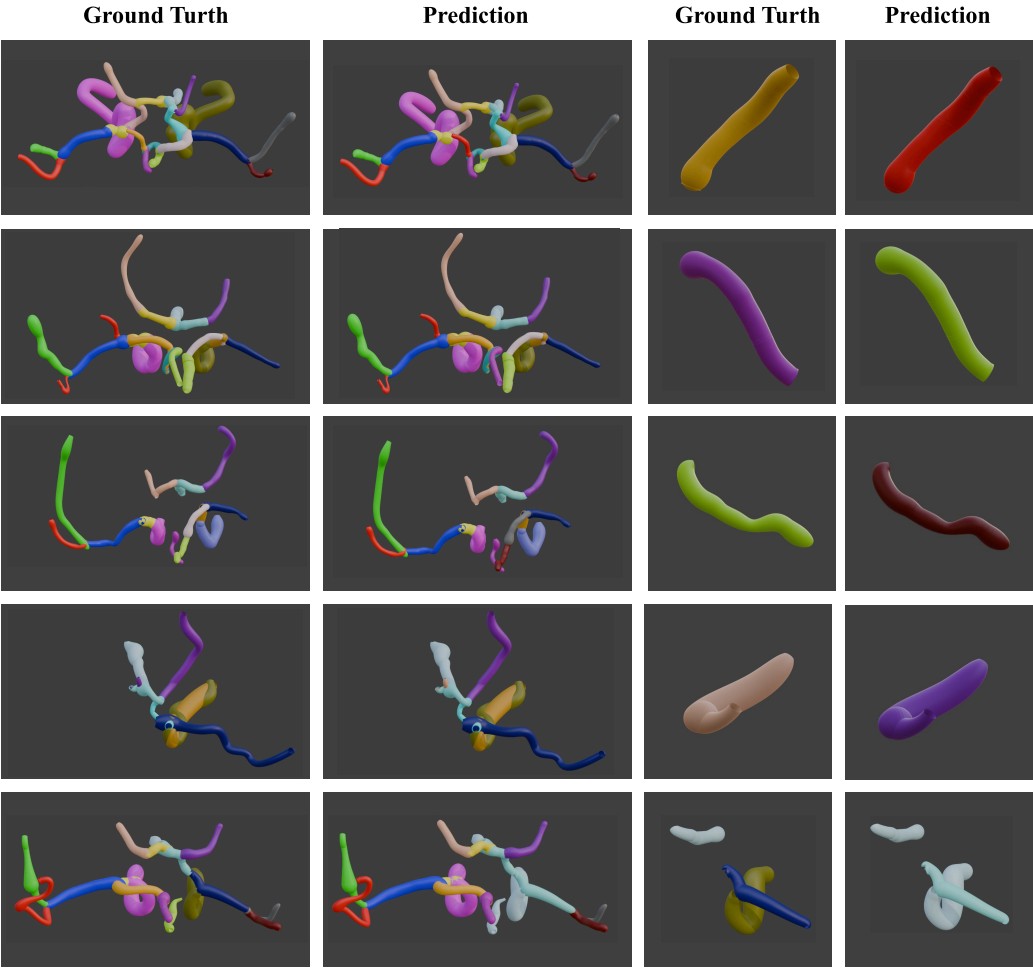

Figure 11: Clinical CoW samples with topological heterogeneity and the corresponding mispredicted structures by SE(3)-BBSCformerGCN. The first two columns show the complete CoW structures, while the last two columns highlight the misclassified branches.

