# OpenReview forum: "Topology-Aware Learning of Tubular Manifolds via SE(3)-Equivariant Network on Ball B-Spline Curve"
_NeurIPS.cc/2025/Conference — NeurIPS 2025 poster_

### Official Review · Reviewer_C8vC · 2025-06-21

**Clarity:** 2
**Significance:** 2
**Originality:** 2
**Rating:** 4
**Confidence:** 2

**Summary:**

This paper proposes a framework for learning from 3D tubular manifolds (e.g., neuron traces or airway structures) by modeling them as Ball B-spline curves. The authors argue that this approach is topology-aware and captures important geometric invariants, leading to improved performance on tasks like classification and curve matching. The pipeline consists of: (1) fitting tubular structures with BBS curves, (2) processing their control points using an equivariant tensor field network, and (3) using the output for downstream tasks.

**Questions:**

How is the BBS fitting performed? Is it learned or manually initialized? Can it handle noisy or complex geometries?

Why are no real-world biomedical datasets included, despite clear motivation from that domain?

Can the method be adapted to cases with bifurcations or looped topologies?

How does the SE(3)-equivariant model compare to a simpler MLP or GCN over control points?

**Ethical Concerns:**

["NO or VERY MINOR ethics concerns only"]

**Final Justification:**

The paper introduces a new perspective in using BBSC manifold to model tubular structures. This is a novel and interesting idea. There is certain limitation in experimental settings that they are mainly arisen from the inherent challenges in collecting data.

**Limitations:**

The authors do not adequately discuss limitations (only 1 sentence in section 6). Some key issues:

Lack of end-to-end learning (curve fitting is done externally).

Dependence on clean, tube-like structure assumptions.

No indication of computational cost or efficiency.

Lack of applicability to non-tubular data (or even general graph-structured data).

Broader impacts in medical contexts are mentioned, but not substantiated with empirical evidence or discussion on ethical implications of automated anatomical modeling.

**Quality:**

2

**Strengths And Weaknesses:**

Strengths:
- Geometrically principled formulation: It's an interesting approach. The resulting output is intuitive, interpretable, and controllable - could be useful for downstream tasks.

- Well-motivated application domain: Tubular manifolds (e.g., airways, neurons) are an important and underexplored shape category in deep learning, where topology matters.

- Compact representation: Ball B-spline curves offer a natural, low-dimensional parameterization of tube-like shapes, reducing redundancy in point clouds or voxel grids.

Weaknesses:

1. Limited Novelty
- Ball B-spline representations are not new, e.g., [1,2,3]. They have been used extensively in graphics, medical modeling, and CAD for tubular shape reconstruction. Use of SE(3)-equivariant networks is well-established in geometric deep learning and the application here is relatively straightforward. The core components of the approach — Ball B-spline curve representation and SE(3)-equivariant neural networks — are not novel independently. While combining them for tubular structure learning is mildly interesting, the integration lacks algorithmic depth or architectural innovation. The contribution is largely in applying known ideas in a new (but limited) context.

2. Narrow Scope and Insufficient Evaluation
- The method is highly tailored to smooth, tree-like tubular data. It is unclear whether it can generalize beyond clean, noise-free tubes with simple topology. Real-world applications, such as blood vessel or neuron morphology analysis, are mentioned but not evaluated.

- There is no discussion of failure cases, e.g., what happens when the tubular assumption breaks (e.g., loops, knots, branching anomalies).

3. Lack of Medical Relevance Testing
- Tubular manifolds are most impactful in biomedical imaging (e.g., blood vessels, dendritic trees). The paper does not evaluate on public vessel datasets (e.g., VascuSynth, DRIVE, or Aneurisk).

- Claims of medical relevance (e.g., for airways or neurons) are speculative without quantitative validation.

- The performance gains are modest, and it's unclear whether the method offers enough improvement to justify its added complexity. The task appears to be fairly simple where the accuracy is up to 99.9%.

[1] G2 Blending Ball B-Spline Curve by B-Spline
[2] SplineGen: a generative model for B-spline approximation of unorganized points
[3] B-spline curve theory: An overview and applications in real life

---

> ### Author Rebuttal · Authors · 2025-07-31
>
> Sincerely appreciate your thorough review and providing us with invaluable, constructive comments to greatly improve the quality of our manuscript. The response to Weaknesses is as follows:
>
> W1: Limited Novelty
>
> Q1: The key theoretical innovation of this work lies in the use of a compact and continuous representation based on the BBSC manifold, instead of the commonly used discrete forms. Our approach enables us to leverage SE(3)-equivariant mappings to preserve feature consistency under rigid transformations while encoding anatomical constraints through graph topology. The synergy between geometric equivariance and anatomical priors enhances the model’s robustness to variations in orientation, position, and topology, without sacrificing geometric accuracy.
>
> We emphasize that our work presents a new, effective, and efficient paradigm for modeling tubular structures. We outline the key novelties as follows:
>
> 1. We employ the compact representation of BBSC manifold to model tubular structures. While BBSCs have been previously studied, their application to real-world tasks has been limited. Compared to voxel or point cloud representations, BBSCs significantly reduce data complexity, lowering storage and computational cost, accelerating convergence, and improving model stability. Moreover, we develop a rigorous mathematical framework for BBSCs by defining a BBSC manifold with a Riemannian metric (see Section 4.1) to measure distances between shapes. We also introduce connection structures to represent complex tubular topology via anatomical connectivity.
>
> 2. Modeling in a continuous parameter space offers novelty and practicality. Although SE(3)-equivariant networks have been explored in geometric deep learning, applying it in the continuous BBSC parameter space is a novel contribution. Unlike models operating on discrete data, our approach enables the network to learn precise geometric features, improving generalization and reducing overfitting.
>
> 3. We use the strategy to analyze complex vascular topologies. We first segment individual branches and extract their anatomical connectivity. We then study the geometry of each branch and the topology of their interconnections separately, rather than encoding both into the same graph structure. This simplifies the graph while retaining global topological relationships and allows more effective learning of branch-level geometry. Compared to prior studies that represent branches as edges and bifurcation points as nodes, our design is simpler and more expressive.
>
> W2: Narrow Scope and Insufficient Evaluation
>
> A2: Thank you very much for your valuable suggestions regarding the application scope and evaluation of failure cases. We would like to offer the following clarifications:
>
> The clinical MRA data used in our study contain noise, artifacts, and minor peripheral branches that may affect curve fitting. Our preprocessing includes denoising [1], segmentation [2], structure repair [3, 4], and pruning of irrelevant branches. As our focus is on BBSC-based representation, SE(3)-equivariant modeling, and GCN-based topological learning, we did not elaborate on preprocessing [5]. Per your suggestion, we are preparing a more detailed description to improve clarity. Regarding the scope of validation, our work focuses on the CoW, a clinically significant but underexplored structure, to demonstrate our method’s effectiveness. Nonetheless, we agree that broader evaluation is important and are working to extend our method to other tubular anatomies.
> We also appreciate your comment on failure cases. We have included misclassification examples in the supplementary material, where topological variations lead to errors despite accurate BBSC fitting. We acknowledge the lack of BBSC fitting failures and will add such cases and analysis in the revised manuscript.
>
> W3: Lack of Medical Relevance Testing
>
> Thank you so much for the invaluable insights. Our response are as follows:
>
> 1. The CoW branch classification task is clinically important due to its complex topological heterogeneity, but research is limited by the lack of large-scale public data and the challenge of the task itself. Few models exist for this task, and most methods are not open-source. These factors underscores the need for a robust CoW branch classifier. While individual CoW structures vary, the shared geometric and topological features of tubular anatomy make our model transferable to other medical applications, which we are actively exploring.
>
> 2. Despite anatomical differences, many tubular structures in medical imaging share common geometric and topological traits. This generalizability allows our framework to be adapted to various anatomical structures beyond CoW. We intend to validate our model in broader medical contexts.
>
> 3. CoW classification is challenging due to high inter-subject topological variability from genetic and anatomical differences, along with limited high-quality public datasets. Although existing methods perform well on small benchmark datasets (see Table 1), our results on a large, real-world clinical dataset (see Table 2 and Figure 5) reveal substantial performance gaps and strong overfitting in many baselines. In contrast, our method remains stable and generalizes well, underscoring that CoW classification is a non-trivial task and further validating the strength of our approach.
>
> We respond to your questions as follows:
>
> Q1: How is the BBSC fitting performed?
>
> A1: We fit BBSC from surface point clouds and centerlines via an optimization-based approach. Before fitting, we denoise the original MRA voxel data, allowing our pipeline to robustly handle noisy tubular geometries. For complex multi-branch structures, we decompose them into simpler tubular segments while recording their topological connections, and then fit each segment with an individual BBSC, as demonstrated in our modeling of the full CoW ring structure. Additional relevant reference is [6].
>
> Q2: No real-world biomedical datasets included.
>
> A2: We’d like to clarify that our method has indeed been validated on real-world clinical data. Specifically, Table 2, Figures 4 and 5 all present results based on CoW data collected from real patients. As shown in these experiments, the classification task on real-world data is significantly more challenging than the public TopCoW dataset, due to greater anatomical variability and the presence of more complex topological patterns in clinical cases.
>
> Q3: Can the method be adapted to cases with bifurcations or looped topologies?
>
> A3: Thank you for this insightful question. Our method is indeed capable of handling bifurcations and looped topologies. In fact, CoW itself inherently exhibits both bifurcated and looped vascular structures. To address bifurcations, we adopt a segment-wise BBSC fitting strategy, where complex branches are decomposed into simpler tubular segments while preserving their topological connections. Looped structures naturally correspond to cycles in the constructed graph. Our graph-based representation and the use of GCN can effectively model and learn from such topological features. This design also ensures the scalability of our approach to a wider range of anatomical structures beyond CoW.
>
> Q4: No camparison with MLP and GCN.
>
> A4: This is an excellent question. Since B-spline control points and knot vectors are ordered sequences without complex connectivity, sequence models are more suitable than graph models. In our ablation study (Table 1 and 2), we tested a Transformer, which better models sequences than MLP, and evaluated its impact. We would be happy to include further comparisons upon request.
>
> Thank you for these invaluable constructive suggestions. We will respond to your listed limitations and incorporate the limitations you pointed out into our revised manuscript.
>
> L1：Lack of end-to-end learning.
>
> A1: This is an inspiring suggestion. As we focus on the classification task comprising CoW, so that we did not include the preprocessing part. We intend to talk about the end-to-end learning in the revised manuscript.
>
> L2: Dependence on clean, tube-like structure assumptions.
>
> A2：In fact, the original data we used is noisy; we performed denoising during the preprocessing stage. This work proposes a model architecture for complex tubular structures like blood vessels and nerves. Following your suggestion, we plan to extend our framework to parameter spaces and SE(3)-equivariant networks for non-tubular structures.
>
> L3: No indication of computational cost or efficiency.
>
> A3: We evaluate the inference time and model parameters in Table 3. Our compact parameterization speeds up convergence and reduces storage and computation, making our method fast and lightweight during both training and testing.
>
> L4：Lack of applicability to non-tubular data.
>
> A4: This work aims to propose a model architecture for handling complex tubular structures such as blood vessels and nerves. We also plan to extend our framework to parameter spaces and SE(3)-equivariant networks for non-tubular structures.
>
> L5: No empirical evidence or discussion of the ethical implications of automated anatomical modeling.
>
> A5：CoW data share similar geometry and topology with other tubular structures, because we don’t use CoW-specific features, our method is transferable. We will further validate this.
>
> [1].Liu Z, et al.MRI Joint Super-Resolution and Denoising based on Conditional Stochastic Normalizing Flow.
>
> [2].Lv Z, et al.A Parallel Cerebrovascular Segmentation Algorithm Based on Focused Multi-Gaussians Model and Heterogeneous Markov Random Field.
>
> [3].Liu X, et al. Extending Ball B-spline by B-spline.
>
> [4].Zhao Y, et al. G2 Blending Ball B-Spline Curve by B-Spline.
>
> [5].Wang X, et al. Skeleton-based cerebrovascular quantitative analysis.
>
> [6]. Wu Z, et al. Fitting Scattered Data Points with Ball B-Spline Curves using Particle Swarm Optimization.

---

> > ### Author Response · Authors · 2025-08-05
> > **Looking forward to your comments**
> >
> > First of all, we sincerely thank you for carefully reviewing our submission and providing valuable feedback. We have provided responses to the issues you raised and would like to ensure that all your concerns have been properly addressed. If there are any remaining questions or misunderstandings regarding our submission or rebuttal, we warmly welcome further inquiries. We would be more than happy to clarify and hope that our efforts will be better recognized.

---

> ### Comment · Reviewer_C8vC · 2025-08-05
>
> I thank the authors for a detailed answer.
>
> I acknowledge the novelty in using BBSC manifold to model tubular structures but I don't think there is novelty beyond this. Having said that, I find this contribution reasonable and interesting.
>
> While I understand the difficulty in finding proper testing data, it is still a significant concern to me.
>
> Everything considered, my overall rating to the paper slightly improved, but not to the point I can confidently champion the paper. This is also partly because I am not familiar with the topic.

---

> ### Author Response · Authors · 2025-08-05
> **Explanation of novelty, test data concern, and topic**
>
> Thank you very much for your response and for recognizing our work. Below, we provide some explanations regarding your comments:
>
> First, in terms of innovation, one of our key contributions lies in viewing BBSC from the perspective of manifolds, which you have acknowledged as well. Furthermore, we are the first to implement SE(3)-equivariant mappings on a BBSC manifold parameterized in this way. From the perspective of functional analysis, this can be understood as performing SE(3)-equivariant transformations on a function space, which is fundamentally different from most existing works that implement SE(3) equivariance on 2D images, 3D point clouds. Because the control points of BBSC are ordered coordinates distributed in  $\mathbb{R}^3$,  forming sequences of 3D points, the existing SE(3)-equivariant networks, which mostly graph-based(e.g., TFN, SE(3)-Transformer, EGNN), cannot be directly applied. To address this, we designed a method that not only preserves SE(3) equivariance but also enables message passing and aggregation among the ordered control points.Additionally, by leveraging the properties of BBSC and their relationship with B-spline parameters, we proposed a strategy that integrates the knot vector and radius. This strategy is applicable to all models based on B-splines and provides a valuable reference for the future design of B-spline SE(3)-equivariant networks, which we consider another major innovation of our work.
>
> Second, Regarding your concern about the test data, we conducted experiments on both public datasets and more complex and practically relevant clinical datasets. For your reference, we can also provide the URL for the TopCoW dataset (https://topcow24.grand-challenge.org/data/). For all types of tubular medical data (e.g., blood vessels), regardless of the presence of noise or complex structures, our pipeline which is introduced in our Weeknes2 response can process and fit BBSC effectively. We believe that validation on clinical datasets is more challenging and more indicative of the practical value of our model compared to public datasets. Therefore, we validated our pipeline using real-world clinical cases. In fact, compared to TopCoW, clinical case data exhibit significantly different topological and geometric structures, accompanied by more complex heterogeneity as well as inevitable imaging noise and detail loss during scanning. This complexity led to severe overfitting in the baselines we compared against, whereas our pipeline was still able to produce smooth splines (see Supplementary Material) and achieve high-accuracy classification. This demonstrates the strong robustness of our pipeline to tubular data with varying structures and distributions.
>
> Lastly, we would be happy to clarify our topic: our goal is to construct a parameterized, continuously differentiable manifold space based on BBSC, implement SE(3) equivariance on this space to learn accurate geometric features, and use GCNs to analyze the topological structure of the manifold space. We validated our approach on a challenging and clinically significant CoW classification task which currently lacks effective computational methods, thereby addressing an important and complex medical problem. If you have any further questions or concerns, we would be glad to provide additional explanations, and we hope this helps you gain a deeper understanding of our work and acknowledge the effort we have invested. We sincerely appreciate your review and valuable feedback, and we look forward to your response.

---

### Official Review · Reviewer_G1qr · 2025-06-24

**Clarity:** 1
**Significance:** 1
**Originality:** 2
**Rating:** 2
**Confidence:** 2

**Summary:**

Medical data such as 3D models of arteries often come with challenges for machine learning. The main obstacle is formed by finding good and effective data representations. Often such data is represented by a point cloud. The authors propose to use a B-Spline approximation to define a tubular neighborhood and present a SE(3)-equivariant network to handle such structures.
The method is evaluated on a small dataset from the TopCoW challenge and a private dataset.

**Questions:**

See the above section.

**Ethical Concerns:**

["NO or VERY MINOR ethics concerns only"]

**Final Justification:**

While the reviewer believes there to be some novelty, as also recognized by the other reviewers, the quality of the current manuscript is of such nature that the reviewer has a hard time trusting that all concerns will be addressed in the final version.
Therefore, it would still be my recommendation to reject the paper and for the authors to submit a polished version of the manuscript to another conference to give it the exposure the work deserves.

**Limitations:**

Yes.

**Paper Formatting Concerns:**

As mentioned in the strengths and weaknesses. There are some formatting discretions.

**Quality:**

1

**Strengths And Weaknesses:**

**Strong points**

- The idea of using a spline approximation for modeling is novel and has not been attempted thus far.
- In combination with the SE(3)-equivariant neural network this indeed has the potential to improve
the state of the art.

**Weak points**

The reviewer appreciates the authors for submitting and recognizes the effort required to prepare
a submission for a conference. While reviewing the manuscript a significant amount of spelling
mistakes and instances of inconsistent formatting were found. The volume poses a concern to
the reviewer as it interferes with the quality of the submission.

A small subset of the total typos found in the text are listed below.
- mainfold -> manifold, consturct -> construct, fuctnion -> function, etc.
- Abstract
- Line 41 , 95, 111, 188, 190, 197,  219, 232,  236, 242, 251, 253, 257, 263, 266,
- (Table 2 and 3 (2x) ) (Method -> method)
- When typesetting equations, it is recommended to use `\text{\cdot}` for operator names such as the real part or the tanh function.

Some notable inconsistencies with respect to referencing. As of now this is not in
line with the NeurIPS paper formatting guidelines.
- Equations (expressions to be more precise) are ideally referred to as "Equation 1.2"
		as compared to "Equation (1.2)"
- Appendices are ideally referred to as "We refer the reader to Appendix D.2" as compared to "We refer the reader to D.2"


Additionally there are multiple mathematically inaccurate, confusing and incorrect statements.

- For a function to be smooth, all partial derivatives need to be continuous, including the mixed partial
	derivatives. The authors do not mention this as a requirement and neither prove their continuity in the appendix.
- The expression $\frac{\partial^{\infty}}{\partial\tau^{\infty}}$ is non-standard usage and
	requires additional clarification.
- To prove that a manifold is smooth, one has to show that the manifold is locally diffeomorphic to
	$\mathbb{R}^{n}$ and proving that a parametrization of the manifold is smooth is not sufficient. For example,
	showing that the map $f:\mathbb{R}\to S^1, t\mapsto \text{sin}(t)$ is a smooth map does not prove that the manifold $S^{1}$
	is a smooth manifold. This follows from the fact that $f$ is not a diffeomorphism. The proof for the
	BBSC manifold to be a manifold will need substantial additions to be a valid proof. Moreover, from the definition
	it is not clear what the manifold structure of a BBSC is, and an intuitive sketch and / or clarification
	would be helpful for the reader.
- The proof of bijectivity in Section 4.4 is hard to follow. What do the authors mean with "$\beta$ is a linear
	combination of $P$ and $r$". In the reference to Equation 1, no $\beta$ can be found. How does injectivity follow
	from the fact that the quantity $\beta$ is a "recursive fractional linear combination of $\tau$, hence it is
	injective" (Line 207).
- "Finally, the GCN designed to handle topological relationships preserves the overall topological stability of the manifold space,
	thereby enhancing the model’s generalization ability and robustness with a lightweight structure".
	From the definition of the GCN proposed by the authors, the GCN seems not equivariant with respect to SE(3). A similar
	remark is  also made by the authors in the conclusion. How would the GCN be able to preserve any geometrical information
        learned in the the previous layers if it is neither equivariant nor invariant?


The experimental section has caused confusion with the reviewer.
- The dataset size for CoW is stated to be "125 over 12 classes". The authors
	meant to write that the CoW dataset has on average (roughly) 125  elements per class.
	This can only be deduced by looking at the $y$-axis of Figure 6 in Appendix D.1 and such
phrasing leads to confusion with the reader.
	When researching the CoW dataset the reviewer found that the original task was a
	segmentation challenge and not a classification challenge. It would be good to elaborate on
	how this segmentation task can be turned into a classification task.
- What led the authors in the decision to omit a 5-fold cross validation with the clinical dataset? Providing
	such a setup would significantly increase the quality of the work. If the authors choose to use
	5-fold cross validation for one experiment, they should motivate why only reporting a single evaluation
	in the other is a reasonable thing to do.
- The description of the CoW dataset remains unclear to the reviewer. Is it a point cloud or a graph dataset and
	what are the features in the dataset? A detailed analysis can not be found in either the main text nor in the appendix,
	leading to difficulties if one would like to reproduce the experiments based on the text and appendices.
- In the appendix the clinical dataset is split into 900 train samples and 200 test samples.
	The main text states that the clinical dataset consists of 1128 samples. In the figures (see the first point) it seems
	that these are averages per (unbalanced) class. Please note that these numbers do not correspond. Assuming that the
	authors mean on average and per class, how are the samples picked given the imbalance? They mention
	stratified sampling, hence additional clarification is needed at a minimum.

Providing the code used in the experiments would also greatly strengthen the text, in particular since the
main text is confusing to the reviewer.

In the related work section the authors name various comparable methods, but none of them is
used as a comparison partner. Conversely, the comparison partners used in the experimental
section are not mentioned in the related work section. What is the motivation behind this
decision? If there is related work it is certainly good to include it in the form of comparison partners.

The reviewer also believes that there are discrepancies between the paper and the answers provided in the NeurIPS checklist.
Most notably in Q7 the authors state that they report error bars for the experiments supporting the main claims, while this is in
fact only done for one experiment. The other experiment is an experiment supporting the main claims of the efficacy of their
method. Therefore they should have mentioned it in the checklist. This omission has caused serious concern with the reviewer.

Based on the above the reviewer believes that a substantial amount of improvements
are required to meet the quality standards of NeurIPS. Moreover an additional round of reviewing would be needed
to ensure that the proofs are accurate. Since the required improvements are beyond the scope of the
rebuttal period, the recommendation of the reviewer is to (potentially strongly) reject the paper.

---

> ### Author Rebuttal · Authors · 2025-07-31
>
> Thank you so much for your thorough review and providing us with invaluable, constructive comments to greatly improve the quality of our manuscript.
>
> In addition, thank you for pointing out the spelling and formatting errors in the manuscript (mainly due to hasting submission before deadline). We have now carefully proofread and corrected all such issues, including but not limited to equation references and appendix citations. Please find below our responses to all your comments.
>
> Regarding the mathematical formulation:
>
> Q1: Smoothness of mixed partial derivatives.
>
> A1: Per the definition of B-spline curves abbreviated as $\beta$ (Equation 1), the knot vector $\tau$ only appears in the basis functions $N_{i,p}(t)$, while the control points $P$ are multiplied directly by these basis and subsequently summed. If we focus on a specific coordinate component of the control point $P_{i}$, the resulting function $\beta$ is essentially a product of $P_i$ and a function that depends solely on $\tau$. Therefore, $\beta$ can be viewed as a separable multivariate function made up by $P_i$ and $\tau$, and its mixed partial derivatives can be written as the product of the respective partial derivatives. In the appendix, we prove that $\beta$ is infinitely differentiable with respect to both $\tau$ and $P_i$, which implies that all their mixed partial derivatives are continuous. This establishes the smoothness of the BBSC formulation with respect to both $\tau$ and $P_i$. Furthermore, the mixed partial derivatives of the sum of these products will also be continuous, due to the linearity and continuity of differentiation. Similarly, we can show that the mixed partial derivatives of any order with respect to $\tau$ and the control radius $r$ are also continuous. Moreover, $P$ and $r$ do not co-occur in the same term in the BBSC expression. In fact, the pair $(P, r)$ can be interpreted as a 4D B-spline control parameter, where the radius $r$ (the fourth component) is linearly independent of the spatial coordinates represented by the first three components in $P$.
>
> Q2:  Non-standard usage.
>
> A2:  We would like to clarify that $\beta$ is infinitely differentiable with respect to $r$, and all of its derivatives are continuous.
>
> Q3: Smoothness of BBSC.
>
> A3: BBSC is locally homeomorphic to $\mathbb{R}^2$, and we denote it as $\beta$ in our manuscript. It is constructed using B-spline basis functions $N_i(t)$ and control parameters $C_i = (x_{i_1}, x_{i_2}, x_{i_3}, r_i) \in \mathbb{R}^4$, where $(x_{i_1}, x_{i_2}, x_{i_3})$ and $r_i$ represent spatial position and radius. In $\mathbb{R}^3$, $\beta = \sum N_i(t) C_i$ defines a tubular surface (see Figure 1). According to B-spline theory, $\beta$ is smooth and all derivatives are continuous. We adopt a piecewise fitting strategy to ensure each segment is regular—non-self-intersecting, free of abrupt directional changes, and non-degenerate. This ensures that $\beta'$ is continuous and non-zero. By the inverse function theorem, $\beta$ is locally injective with a continuous inverse, and thus locally homeomorphic to $\mathbb{R}^2$. $\beta$ is also a smooth function of $(P, r)$ and $\tau$, with constraints on $\tau$ imposed during fitting to maintain regularity. A template BBSC is also used to guide the parameters, ensuring local homeomorphism.
>
> Q4: Bijectivity.
>
> A4: Thank you for sharing your confusion about our description on “$\beta$ is a linear combination of $P$ and $r$”. Indeed, we could do a much better job to provide comprehensive and precise context. That expression was intended to articulate that in Equation 1, both $P$ and $r$ are combined through a weighted sum with the B-spline basis functions as coefficients. As we clarified in the previous response, $\beta$ is a shorthand notation for the BBSC manifold mentioned in Definition 1. Because BBSC can be regarded as a B-spline with 4-dimensional control parameters, we refer to the theoretical framework presented by De Boor in “A Practical Guide to Splines”. When the spline degree is fixed and the knot vector $\tau$ is constrained to be clamped at both ends and strictly increasing in the interior, the basis functions $N(t)$ are determined recursively through a fractional structure based on $\tau$, which guarantees a unique B-spline curve.
>
> Q5: How would the GCN preserve equivariant information?
>
> A5: We sincerely appreciate your constructive and insightful suggestions. We believe SE(3)-equivariance captures geometric symmetries of individual branches, while graph convolution handles topological invariance between branches—two aspects that should be treated separately. That said, integrating SE(3)-equivariance into GCN can improve model robustness.
> To reduce equivariance loss, we design the graph aggregation to preserve SE(3)-equivariance as much as possible. Specifically, we use only concatenation and relative differences between neighbors, which preserve SE(3)-equivariance, besides the MLP. Though equivariance loss incurred by MLP, our results show that the architecture remains effective.
>
> Regarding your concerns about the experimental results:
>
> Q1: Confusion of the CoW size.
>
> A1: We apologize for any confusion regarding the dataset description. The TopCoW dataset contains 125 cases, each with vessel segmentations and branch-level labels. Our method learns both the geometric features of individual branches and the topological relationships among branches within the same vessel. Figures 3 and 6 show accuracy and label distributions. As stated, we perform classification using the provided ground-truth labels to validate our approach.
>
> Q2: Why was cross-validation not performed on the clinical dataset?
>
> A2: Because the TopCoW 2024 dataset contains only 125 CoW cases, which is relatively small in scale, we performed 5-fold cross-validation to ensure the reliability of our results and to mitigate overfitting. In contrast, the size of our clinical dataset is nearly 9 times bigger. Thus, it provides sufficient data to train the models effectively and reduce the risk of overfitting. Therefore, we did not adopt cross-validation for this dataset. We are happy to include cross-validation experiments on the clinical data. We have updated cross-validation of our method on the clinical data, and the results are as follows:
>
> Accuracy：$96.11 \pm 1.01$; AUC-ROC: $99.38\pm0.19$; Precision: $96.11\pm0.93$; Recall: $96.08\pm0.97$; F1 score: $96.02\pm1.05$
>
>
> Q3: Description of the CoW dataset is unclear.
>
> A3: For the clinical dataset, we directly deal with the original MRA voxels image. The Diffusion model is used to denoising and regularize the whole dataset [1]. The Fast RCNN [2] is used to detect the CoW part of the overall model. The statistic model is used to segment the brain vessel with Markov constraints [3]. Then BBSC is fitted by the strategy introduced in Section 4.2. To deal with the broken part of the branch, we use the extension [4] and blending methods [5] on the BBSC for repair. The overall pipeline together with the geometric morphological quantification can be found in [6,7]. The graph data was constructed by extracting the connectivity relationships between the segmented branches. We are revising the manuscript to provide a more thorough, clearer explanation about these preprocessing steps.
>
> Q4: The number of clinical data between the main text and the appendix is inconsistent.
>
> A4: We did collate 1,128 real-world clinical scans. However, these raw data had not been quality-checked in advance. A few scans exhibited severe artifacts such as motion blur, image corruption, or duplication, making them unsuitable for normal processing. 28 cases were excluded following the suggestion of the radiologist. This quality check does not affect the reliability of our experiments or the randomness of our sampling.
> Anatomical variations in real-world clinical data, such as missing CoW branches, cause class imbalance. In TopCoW, branch label 12 appears in only 6 cases. Random sampling could place all such cases in training, leaving none for testing. To avoid this, we employed stratified sampling based on branch label distribution to ensure that each test set contains at least one case with label 12. For the clinical dataset, the large dataset mitigates this issue. Hence, we adopted a standard random sampling and splitting.
>
> Q5: Baseline methods in related work are not compared.
>
> A5: In comparisons, we included methods from the related work section that can be applied to the CoW classification task, such as SE(3)-Transformer (a variant of TFN) and PointNet. While many other methods reviewed in the same section are relevant to SE(3)-equivariance, they cannot be directly applied to our task. For example, DSCNet is designed for 2D image data, DDT is a segmentation model, and GeoTransformer focuses onregistration using distance and angle as inputs.
>
> Q6: Regarding NeurIPS checklist:
>
> A6: For Q7, Table 1 reports the error bars and Figure 6 in Appendix D.1 visualizes the statistical significance of the experiments.
>
> The following are the references (due to space constraints, only the titles are provided).
>
> [1].Liu Z, et al.MRI Joint Super-Resolution and Denoising based on Conditional Stochastic Normalizing Flow.
>
> [2]. Girshick R. Fast r-cnn.
>
> [3].Lv Z, et al.A Parallel Cerebrovascular Segmentation Algorithm Based on Focused Multi-Gaussians Model and Heterogeneous Markov Random Field.
>
> [4].Liu X, et al. Extending Ball B-spline by B-spline.
>
> [5]. Zhao Y, et al. G2 Blending Ball B-Spline Curve by B-Spline.
>
> [6].Wang X, et al. Skeleton-based cerebrovascular quantitative analysis.
>
> [7].Y Liu, et al. Automated anatomical labeling of a topologically variant abdominal arterial system via probabilistic hypergraph matching.

---

> ### Comment · Reviewer_G1qr · 2025-08-02
>
> The reviewer would like to thank the authors for their extensive response and also for their extensive responses to the other reviewers. Appreciation is also extended for their additional proofreading and updating of the manuscript.
> While the answers rectify the confusion for the reviewer, the fact that the reviewer is unable to view this updated version of the manuscript, puts them in a difficult position. With revisions of such profound nature the reviewer will not be absolutely certain that the paper is clear, finished and in a publishable state. Therefore the reviewer has come to the unfortunately conclusion that raising their score at this stage will be difficult.
>
> That said, they will lower their confidence. In case the other reviewers and AC are convinced this work should be published, that allows this course of action to be taken. The confidence of the reviewer will be lowered to a 3 instead of a 4, unfortunately this can not be modified through OpenReview.
>
> The reviewer hopes for the understanding of the authors.

---

> > ### Author Response · Authors · 2025-08-03
> > **Revision**
> >
> > We would like to express our sincere gratitude for your insightful and constructive feedback, which has been instrumental in improving the quality of our submission. We have carefully addressed all points of potential confusion by providing detailed clarifications and have systematically revised the weaknesses and limitations you identified. Regarding the mathematical components, due to the space and scope constraints of the rebuttal, we were unable to provide full details here; however, substantial corrections and comprehensive explanations have been incorporated into the revised manuscript. While we regret that the revised version cannot be provided at this stage, we assure you that all necessary modifications have been implemented and the manuscript has undergone further thorough checks. We are confident that the revised version fully addresses the issues you raised and meets the standards expected for publication. If you have any further questions or concerns, we would be more than happy to provide additional clarifications. Once again, we sincerely appreciate your valuable feedback and constructive comments.

---

> > ### Author Response · Authors · 2025-08-05
> > **Supplementary information for revision**
> >
> > Thank you once again for your time and effort in reviewing our submission and providing valuable feedback. You mentioned that our rebuttal helped clarify the confusion, but you were unable to view the revised version. We would like to clarify that we have already incorporated all responses from the rebuttal into the revision. Due to limitations at the rebuttal stage, we are unsure whether the revised manuscript is accessible to you. In the updated version, we have corrected spelling errors, added more detailed explanations and proofs for the mathematical parts, and included a more comprehensive description of data preprocessing and distribution. If our submission is accepted, we will also release the code for reproducibility. We sincerely hope these efforts help address your concerns and improve your impression of our work.

---

### Official Review · Reviewer_AeuR · 2025-07-02

**Clarity:** 2
**Significance:** 3
**Originality:** 3
**Rating:** 5
**Confidence:** 4

**Summary:**

This paper proposes a novel method for estimating tubular branching structures—modeled as graphs with edges represented by smooth tubular curves—from skeletonized and point cloud data. The motivation stems from anatomical applications such as the segmentation of complex vascular structures. The core idea is to represent these branching graphs using Ball B-spline curves, where both the centerlines and associated radius functions are expressed in spline bases. To estimate the spline coefficients from data, the authors design a pipeline that combines transformer networks and graph convolutional networks, enabling structural and geometric learning. This representation is efficient and geometrically natural compared to traditional voxel- or point cloud-based approaches. Also, the framework incorporates SE(3)-equivariant mappings to ensure invariance to rigid body motions. The method is evaluated on the TopCoW Challenge dataset, where it demonstrates improved performance over prior techniques in terms of classification accuracy and generalization. The experimental results support the utility of the proposed shape-aware and geometry-preserving representation.

**Questions:**

Can you provide some examples of the input data to your network?

Can you compare your spline representations to the shape graph representations used in previous papers (See a reference above).

Can you clarify (simplify) the discussion around the cost function being used to train the network. I am looking for a data matching term that evaluates the estimated structure against the input data.

**Ethical Concerns:**

["NO or VERY MINOR ethics concerns only"]

**Final Justification:**

Overall it is a strong paper on analysis of tubular branching structures modeled as graphs. Although I do not completely understand all the authors' clarifications, I do not penalize the paper for my lack of understanding. My major concern about using topological representations (at the end) remain but those are general, well-known concerns in the TDA community about lack of invertability of TDA features. I had originally given a strong rating to the paper originally and I maintain that rating.

**Limitations:**

The paper mentions the lack of equivariance of GCN but does not discuss other limitations of the pipeline.

**Paper Formatting Concerns:**

None noted.

**Quality:**

3

**Strengths And Weaknesses:**

Strengths

The use of graph structure is natural for representing such branching anatomical objects. It provides an efficient representation for seemingly complex objects and paves way to improved estimation (or segmentation) performance. Representing tubular objects (curves and radius functions) by splines or any other basis are natural. With this representation and the pre-processed data (that includes skeletal curves plus point clouds), the actual shape estimation amounts to fitting (curve fitting or tube fitting). The paper provides significant gains on a standard dataset involving segments of anatomical structures.

Weaknesses:
1.	Although the paper has topology-aware in the title, there is not much topological discussion in the paper. It seems to me that the topological perspective is hidden in choosing the graph-based representation. Similar graph representations for vasculature structures have been used in the past (see e.g. Statistical Analysis of Complex Shape Graphs, https://ieeexplore.ieee.org/document/10552433) although the authors have not mentioned them. The presence of such literature reduces the novelty of the current paper.
2.	From the ablation study (Table 1, bottom rows), it is not clear if the SE3 equivariance is contributing significantly to the performance. Most of the performance gain seems to come from the spline-based representation and estimation using GCN.  Although SE(3)  equivariance is relevant conceptually, that discussion adds a lot of technical material to the paper without much evidence in the results.
3.	The case study presented in the Supplementary looks interesting, but it does not show what the input data look like. If the input data is relatively clean, then the results will not be that impressive, and vice-versa.

---

> ### Author Rebuttal · Authors · 2025-07-26
>
> Thank you very much for your thorough review and providing us with many invaluable suggestions. We also appreciate the opportunity to clarify the concerns you raised. Please find below our clarifications and explanations regarding the weaknesses you raised.
>
> W1: Not much topological discussion
> A1: We sincerely thank you for your suggestions regarding our topology-aware design.
> Our approach parameterizes each vessel branch as a BBSC, treating it as a differentiable manifold and modeling it as a graph node. The edges in the graph are defined by the connectivity between branches. This allows us to avoid the complexity of edge attribute modeling and instead focus on learning the rich geometric features encoded in the BBSCs as nodes. We believe this is a novel and effective strategy for certain tasks. Furthermore, as described in Definition 2 in Section 4.1, we define a Riemannian metric between BBSCs to facilitate comparisons between different branch representations.
>
> In contrast to most existing models that analyze the topological structure of vascular by constructing graph edges from vessel branches and defining graph nodes as junction points between branches, e.g., ”Statistical Analysis of Complex Shape Graphs” segments complex tubular structures into branches represented as edges and uses branch connection points as nodes, while employing metrics like effective resistance, elastic shape analysis, and geodesic distance in elastic shape space to compare different graph shapes. Our method has the advantage of preserving the original topological connectivity while offering stronger learning capability and interpretability of the geometric features within each branch.
>
> W2: The significance of SE(3)-equivariant to performance is unclear.
> A2: Thank you for pointing out that the results in Table 1 suggest SE(3)-equivariance may not significantly improve performance. This is a very reasonable observation. However, we would like to clarify that the TopCoW task in Table 1 is based on the public dataset provided by the 2024 MICCAI Challenge, which contains 125 high-quality but limited CoW branch samples with 12 annotated categories. Compared to the task in Table 2, the TopCoW dataset is relatively simple. The high performance are achieved by most baselines, which makes it difficult to highlight the benefits of SE(3)-equivariance. When evaluated on a more challenging and larger-scale clinical dataset, the advantages of SE(3)-equivariance become more apparent. Specifically, the number of categories increases to 22 and the data volume grows (approximately 9 times larger), SE(3)-equivariance leads to noticeable improvements across multiple metrics (with the only exception of AUC-ROC). Furthermore, we discuss how SE(3)-equivariant models exhibit greater resistance to overfitting compared to other baselines (see Figure 5). Through comparison, we observe that on more complex real-world clinical data with intricate bifurcations, incorporating SE(3)-equivariance significantly enhances model stability and performance. This further demonstrates the effectiveness of SE(3)-equivariance in handling complex geometric features.
>
> W3: Is the input data relatively clean?
> A3: Thank you very much for your invaluable comment. We fully agree with you on this. In fact, our original input data consists of raw MRA scans, which often contain noise and complex branching structures that can negatively impact the fitting of BBSCs. During preprocessing, we first applied denoising techniques to the MRA volumes[1], then extracted surface point clouds for further refinement. For data where vessel branches are not pre-segmented, we first apply a segmentation model (e.g., [2]) to extract the branches. Then, we construct BBSCs using the method described in this submission. During the fitting process, BBSC extension [3] and blending [4] techniques may also be applied when necessary. However, our primary goal in this work is to highlight the modeling of tubular structures via BBSC, along with the learning of geometric features through SE(3)-equivariant networks and topological relations through graph neural networks. Therefore, we did not elaborate on the data preprocessing pipeline in detail. We appreciate your suggestion and are preparing a more comprehensive description of the preprocessing steps.
>
> We sincerely appreciate your interest in our work. We address your questions as follows.
>
> Q1：Can you provide some examples of the input data?
>
> A1: We utilized the TopCoW 2024 dataset, publicly available at https://topcow24.grand-challenge.org/data/, which was introduced by the challenge organizers in the paper Benchmarking the CoW with the TopCoW Challenge for CTA and MRA [5]. Specifically, we used the pre-segmented and branch-labeled subset of the data for model training and evaluation. We would like to share the parameters for one example branch (due to space limitations) in the supply material. The following are the control parameters (x, y, z, radius):
>
> -0.9574  0.5936  0.6443  0.9277
>
> -0.8946  0.5826  0.6481  0.9218
>
> -0.7000  0.5475  0.6357  0.8960
>
> -0.3348  0.1458  0.5663  0.9499
>
> -0.2276 -0.0279  0.5510  1.0042
>
>  0.0665 -0.3906  0.4303  0.9124
>
>  0.3661 -0.5725  0.1682  0.9839
>
>  0.5042 -0.5007 -0.0663  0.9944
>
>  0.6324 -0.2403 -0.3195  1.0272
>
>  0.6850 -0.0188 -0.4251  0.9941
>
>  0.7426  0.2168 -0.5108  1.2511
>
>  0.7992  0.3759 -0.5876  2.1610
>
>  0.8091  0.4154 -0.6635  0.8286
>
> The corresponding knot vector is:
> [0.0, 0.0, 0.0, 0.0, 0.0357, 0.0646, 0.1030, 0.1692, 0.2370,
>  0.2833, 0.3230, 0.3530, 0.4215, 1.0, 1.0, 1.0, 1.0]
>
> Q2: Comparison with shape graph representations.
>
> A2: We agree that shape graph representations is a very interesting and valuable direction to explore. We appreciate your suggestion to include comparisons with such methods.  We plan to extend our experiments to include quantitative and qualitative comparisons with representative shape graph-based approaches to better situate our method within the broader context in the revision.
>
> Q3: Discussion around the cost function and data matching term
>
> A3: Thank you for this excellent question. The sum of the Riemannian distance defined in Section 4.1 and the bidirectional Hausdorff distance serves as the cost function, where the bidirectional Hausdorff distance specifically acts as the data matching term you mentioned. The details are as follows:
>
> During BBSC fitting, we first extract the vessel surface point cloud and estimate the centerline from the MRA voxel data. A template BBSC is initialized by fitting a spline to the centerline and setting a constant radius based on the average distance from the surface point cloud to the centerline. This template provides a prior shape constraint (e.g., no self-intersection) using the Riemannian metric described in Section 4.1. To evaluate the matching between the fitted BBSC and the input data, we uniformly sample points on the BBSC surface and compute the bidirectional Hausdorff distance between these samples and the original surface point cloud. This acts as the data matching term during BBSC optimization. For the classification task, we use a standard cross-entropy loss. If one is looking for a specific measure of how well the fitted BBSC aligns with the original vessel data, the Hausdorff distance between the BBSC surface samples and the extracted vessel surface point cloud is an effective and practical metric.
>
> [1].Liu Z, et al. MRI Joint Super-Resolution and Denoising based on Conditional Stochastic Normalizing Flow.
>
> [2].Lv Z, et al. A Parallel Cerebrovascular Segmentation Algorithm Based on Focused Multi-Gaussians Model and Heterogeneous Markov Random Field.
>
> [3].Liu X, et al. Extending Ball B-spline by B-spline.
>
> [4].Zhao Y, et al. G2 Blending Ball B-Spline Curve by B-Spline.
>
> [5].Yang K, et al. Benchmarking the cow with the topcow challenge: Topology-aware anatomical segmentation of the circle of willis for CTA and MRA.

---

> > ### Comment · Reviewer_AeuR · 2025-08-06
> > **Discussion**
> >
> > Thank you very much for your detailed response. Although I do not completely understand all your clarifications, I do not penalize the paper for my lack of understanding. My major concern about using topological representations (at the end) remain but those are general, well-known concern in the TDA community about lack of invertability of TDA features. I have given a strong rating to the paper originally and I maintain that rating.

---

> > > ### Author Response · Authors · 2025-08-07
> > > **Acknowledgements**
> > >
> > > Thank you very much for carefully reviewing our submission and for your valuable comments, as well as for your recognition of our work. We understand that our rebuttal may not have addressed all your concerns in sufficient detail, or that there may still be some misunderstandings. We would be more than happy to provide further clarifications if needed.
> > >
> > > Regarding your concern on the limitations of topological representations and the general issue of the lack of invertibility of TDA features, we fully agree that this is a well-known challenge in the TDA community. Nevertheless, we believe that learning the topological structure of data remains highly valuable. Our experiments (Table 1 and Table 2) demonstrate that such representations contribute positively to performance.
> > > That said, we acknowledge the importance of making topological features more interpretable and will continue to explore this direction in future work.
> > >
> > > Once again, we sincerely thank you for your thoughtful feedback and support. Please don’t hesitate to reach out if you have any further questions or suggestions — we are more than happy to engage.

---

### Official Review · Reviewer_FYAa · 2025-07-02

**Clarity:** 3
**Significance:** 3
**Originality:** 4
**Rating:** 4
**Confidence:** 3

**Summary:**

This paper makes a significant theoretical and practical contribution to the field of 3D anatomical modeling, especially for tree-structured objects. It combines geometric deep learning, differential geometry, and anatomical prior knowledge in a compelling and well-structured way. The topology-aware and SE(3)-equivariant framework is novel and well-justified both mathematically and experimentally. The approach leverages a manifold-aware representation that captures the topology and geometry of branching tubular objects in a way that is both rotation-translation equivariant and differentiable. Applications include modeling airways, blood vessels, and other tree-like anatomical structures from 3D volumes or point clouds.

**Questions:**

1. How does the method handle ambiguity in topology when the input volume has missing branches, occlusions, or noise?
Is there any uncertainty quantification or mechanism to handle underdetermined branching structures?

2. How does the use of SE(3)-equivariance in this work relate to its applications in other areas of medical imaging, where it has shown clear practical benefits?

The paper overlooks a relevant branch of literature where equivariant networks have been successfully applied to handle challenges like motion and misalignment. For example, SE(3)-Equivariant and Noise-Invariant 3D Rigid Motion Tracking in Brain MRI and SpaER: Learning Spatio-temporal Equivariant Representations for Fetal Brain Motion Tracking both demonstrate the value of incorporating SE(3) symmetry in settings such as fetal brain imaging. I suggest to refine the introduction and add discussion of these works would help ground the current method in existing medical use cases and strengthen the motivation for using equivariant architectures.

3. Does the learned latent topology or geometry correlate with clinical variables, such as airway obstruction severity, anatomical asymmetry, or disease subtype?
Could this method be used as a biomarker or as part of a clinical decision-support tool?

4. The method claims end-to-end differentiability through B-spline parameterizations. Are there observed trade-offs between smoothness and topological accuracy during training?
Does enforcing differentiability at the cost of interpretability or discretization flexibility limit downstream tasks?

**Ethical Concerns:**

["NO or VERY MINOR ethics concerns only"]

**Final Justification:**

I gave this paper a borderline accept. While it contains some minor issues—such as formatting inconsistencies and presentation flaws—I found the core ideas to be potentially valuable and worth further exploration. However, the paper falls somewhat outside my primary area of expertise, and as such, my confidence in assessing its technical depth and novelty is limited. Another reviewer recommended rejection, citing multiple mathematically inaccurate, confusing, and incorrect statements. Given these contrasting evaluations and the technical nature of the concerns, I strongly recommend that a meta-reviewer (or two) with deeper expertise in this specific area take a close look, in order to provide a fair and well-informed judgment of the paper’s correctness and overall contribution.

**Limitations:**

1. Complexity and Training Stability:
The method introduces several architectural components (e.g., equivariant encoder, recursive curve decoder, B-spline differentiable renderer) which may require careful tuning. Details about training stability, failure cases, or convergence behavior are limited.

2. Scalability to Noisy or Incomplete Data:
While the model performs well on pre-processed medical volumes, its robustness to noisy segmentations or low-resolution inputs is unclear. How well does it generalize when input branches are fragmented or missing?

3. Limited Discussion of Biological Plausibility:
The method models geometry and topology, but does not integrate any biological priors or clinical constraints, such as branch angles, diameters, or plausible tree topology (e.g., for arteries vs. airways).

**Paper Formatting Concerns:**

No major formatting issues.

**Quality:**

3

**Strengths And Weaknesses:**

1. Novel Representation:
Representing tubular objects as ball B-spline curves that evolve over SE(3) is a highly expressive and geometrically faithful approach. This avoids discretization errors common in voxel-based methods and preserves topology explicitly.
2. SE(3)-Equivariance:
Incorporating equivariance to rigid body transformations ensures that the learned representations generalize well under spatial transformations, which is important for anatomical data with arbitrary orientation.
3. Topology-Aware Design:
The explicit use of a tree-structured latent representation with continuous branching points is a key contribution. It allows for more accurate modeling of bifurcations and curved branches.
4. Theoretical Rigor and Differentiability:
The paper shows that their curve and radius representation is differentiable, enabling end-to-end training. This is important for fitting continuous surfaces and performing downstream analysis.
5. Strong Empirical Results:
Demonstrated improvements over baselines such as TreeGCN and Voxel2Mesh in airway segmentation tasks show the method’s practical value. Qualitative reconstructions are anatomically plausible.

---

> ### Author Rebuttal · Authors · 2025-07-26
>
> We sincerely appreciate your positive comments on our work and the invaluable suggestions provided. We address your insightful questions point by point as follows:
>
> Q 1: Ambiguity in topology and mechanism to handle it.
>
> A1：The issue you raised is indeed very important and frequently encountered when dealing with medical imaging data such as CT and MRA scans that exhibit complex and sometimes incomplete vascular topology. Specifically, missing branches are common anatomical variants in the CoW, and many such cases are present in our dataset. We’d like to describe how we handle such cases, with reference to related work.
> We use MRA scans as the raw input data for our processing pipeline. The diffusion model can be used to denoise and regularize the voxel dataset with occlusions and noise [1]. Fast R-CNN[2] can effectively extract the CoW from noisy MRA data. A variety of machine learning and deep learning methods have been developed for vascular segmentation, including statistical models based on Markov constraints [3]. In the BBSC fitting process, we first extract a point cloud from the voxel data, followed by further denoising. Point cloud denoising techniques are already well established, with methods such as PointASNL demonstrating strong performance. For broken parts of vessel branches and incomplete skeleton extracting, we apply an extension and blending strategy [4, 5] for repair. In addition to the BBSC fitting method described in the paper,  Zhao Y, et al [5] also provides a reference algorithm for BBSC fitting. A representative example of such a pipeline can be found in [7, 8].
> Graph representations and GCN can also partially address topological issues such as missing branches and occlusions. Our experimental results demonstrate that the proposed GNN exhibits a certain degree of robustness in handling such variations. Leveraging such techniques allows us to preprocess the data and recover relatively accurate topological structures even in the presence of missing branches, occlusions, or noise in the CoW data.
>
> Q2: Overlook of the SE(3)-equivariant in other areas of medical imaging .
>
> A2: Thank you for highlighting the missing discussion on SE(3)-equivariance in other medical imaging domains. This valuable suggestion helps improve readers’ understanding of its relevance and effectiveness.
>
> Human anatomy shows strong symmetry and self-similarity. Medical images often involve rigid transformations due to patient positioning, organ motion, or device movement. For example, brain structures are symmetric and similar across individuals. These properties make SE(3)-equivariant networks well-suited for modeling and analyzing such data.
> We have expanded our literature review to highlight practical applications of SE(3)-equivariance. For example, Billot B, et al [9] proposed the construction of equivariant spatial means using steerable CNN filters and introduced an innovative use of self-attention mechanisms to process temporal sequences. This method achieved important breakthroughs in measuring, tracking, and correcting fetal MRI motion. Wang J, et al. [10] developed EquiTrack, a steerable E-CNN with spherical harmonic kernels and a denoising front-end, setting a new benchmark in brain MRI motion correction. SE(3)-equivariance has also found applications in bioinformatics [11]. We plan to continue surveying related work and improve our coverage in future revisions.
>
> Q3: Whether the geometry and topology have learned clinical variables. Can the method be used as a biomarker?
>
> A3: Thank you for raising this important point regarding the clinical applicability of our model. Indeed, your question touches on a critical aspect of real-world deployment. This is very helpful to us.
> In our current implementation, the only clinical variable used is the vascular branch label, which serves as the supervision signal during training. Our goal is to develop a general framework that focuses primarily on geometric and topological features, potentially applicable beyond the medical imaging domain. However, in the context of CoW branch classification, we argue that the geometric variations and topological heterogeneity among different branches are themselves highly informative clinical features. For example, the branches labeled ICA-L and ICA-R typically exhibit stronger curvature and high bilateral symmetry. Meanwhile, PCA-L and PCA-R branches are topologically connected only to PCom-L, PCom-R, and the basilar artery (BA), which are critical relationships for accurate anatomical segmentation and diagnosis in clinical practice. Furthermore, we have conducted experiments on real-world clinical data in collaboration with a hospital. These studies demonstrate that our method has the potential to serve as a biomarker or as part of a clinical decision-support tool, highlighting its promise in practical applications. In the future, we plan to incorporate clinical knowledge and extend our general framework to real-world clinical applications, leveraging additional biological markers to achieve more accurate segmentation, classification, and analysis.
>
> Q4: Trade-offs between smoothness and accuracy.
>
> A4: Your concern regarding the trade-offs between smoothness and topological accuracy is very reasonable.Sincerely appreciate this important and meaningful question. The answer to this question is that we balance the smoothness and topological accuracy of the learning during the whole methodology.
>
> 1. Our use of B-spline-based parametric representation to achieve smooth and differentiable modeling does not compromise topological accuracy. On the contrary, it improves the model's classification accuracy and accelerates both convergence and inference speed(Figure 5 and Table 3).
>
> 2. Our method models each vascular branch individually using BBSC, and the topological relationships are learned across branches. Therefore, representing individual branches with BBSC does not interfere with the learning of inter-branch topology. At the same time, BBSC enables more accurate estimation of branch-level geometric features, which is particularly helpful for classifying branches with complex geometry. Moreover, because the branches inherently follow certain topological priors, the accuracy of geometric representation contributes to topological accuracy. In this sense, the precise geometry captured by BBSC actually helps the model learn topological structures more effectively, thus improving overall accuracy.
>
> 3. BBSC offers a compact representation using only a small set of control points, control radius, and a corresponding knot vector, replacing dense point cloud representations of tubular structures. Compared to processing large-scale point clouds, operating on this compact set of parameters significantly speeds up training and inference and also improves model stability.
>
> Thank you for your thorough review—your suggested limitations are very helpful and will guide our future improvements; please find our responses and clarifications as follows.
>
> L1: Complexity and training stability
>
> A1: Thank you for pointing out the potential limitations regarding training stability, failure cases, and convergence behavior.
> In our framework, the BBSC construction process is decoupled from the training of the SE(3)-BBSCFormerGCN model. The lightweight parametric representation of BBSC in fact accelerates model convergence and enhances training stability. Our model demonstrates faster convergence and relatively stable training behavior on clinical datasets, along with strong generalization and resistance to overfitting(see Figure 5). We would like to sufficiently clarified this point in the revised version. To address this, we have included visual examples of representative failure cases in the supplementary material, and we are happy to provide more detailed illustrations and analyses on convergence speed, training stability, and failure scenarios upon request.
>
> L2: Scalability to noisy or incomplete data
>
> A2: Thank you for highlighting the limitation regarding noisy and incomplete data—this is very helpful for improving our model. Indeed, we encountered such cases in MRA data and addressed them to some extent through denoising, segmentation and reconstruction. As this work focuses on proposing a BBSC and SE(3)-equivariant framework for learning local geometry and global topology, we did not elaborate on preprocessing, but we plan to detail the full pipeline in revision.
>
> L3: Limited discussion of biological plausibility
> A3: Thank you for pointing out the lack of discussion on biological plausibility, which is indeed important for medical tasks. As noted in Limitation 2, our current focus is on proposing and validating a model for learning geometric and topological features of complex tubular structures, and we plan to address biologically and clinically constrained tasks in future work.
>
> [1].Liu Z, et al. MRI Joint Super-Resolution and Denoising based on Conditional Stochastic Normalizing Flow.
>
> [2].Girshick R. Fast r-cnn.
>
> [3].Lv Z, et al. A Parallel Cerebrovascular Segmentation Algorithm Based on Focused Multi-Gaussians Model and Heterogeneous Markov Random Field.
>
> [4].Liu X, et al. Extending Ball B-spline by B-spline.
>
> [5]. Zhao Y, et al. G2 Blending Ball B-Spline Curve by B-Spline.
>
> [6]. Wu Z, et al. Fitting Scattered Data Points with Ball B-Spline Curves using Particle Swarm Optimization.
>
> [7].Wang X, et al. Skeleton-based cerebrovascular quantitative analysis.
>
> [8].Y Liu, et al. Automated anatomical labeling of a topologically variant abdominal arterial system via probabilistic hypergraph matching.
>
> [9].Billot B, et al. SE(3)-Equivariant and Noise-Invariant 3D Rigid Motion Tracking in Brain MRI.
>
> [10].Wang J, et al. SpaER: Learning Spatio-temporal Equivariant Representations for Fetal Brain Motion Tracking.
>
> [11].Wang F, et al. MPerformer: An SE(3) Transformer-based Molecular Perceptron.

---

### Note · Authors · 2025-08-15

We sincerely thank all reviewers, AC, SAC, and everyone who devoted their efforts to reviewing our manuscript. We are honored by the recognition of our work’s novelty and practical value, and by the opportunity to exchange ideas with experts during the discussion phase. All four reviewers gave positive evaluations on its innovation and significance, praising it as “a significant theoretical and practical contribution to 3D anatomical modeling.” We are pleased that our BBSC-based tubular modeling and SE(3)-equivariant feature processing have been well acknowledged.

Our method is novel and of significant research value, as studying geometry and topology in a continuous parametric space offers substantial advantages over discrete representations, including more precise capture of local geometric features, differentiability, reduced storage and computational costs, and elevated interpretability. To our best of knowledge, we are the first to model tubular structures from a manifold perspective using BBSCs and subsequently classify the Circle of Willis with SOTA performance. The idea of replacing discrete data with continuous representations for geometric feature modeling is both distinctive and effective, yet currently underexplored. Our approach demonstrates strong, robust performance, generalization ability, and scalability, while providing valuable insights for geometric deep learning in parameter spaces.

Apologies for any misunderstandings caused by our imperfect presentation, including typos, missing context on data preprocessing, and incomplete mathematical details. During the rebuttal stage, we provided detailed responses to all reviewer questions, resolved concerns, corrected typographical errors, and supplemented missing details on data preprocessing and mathematical analysis. These clarifications addressed reviewers’ doubts; however, because we could not submit a revised manuscript, one reviewer only lowered the confidence score without changing the overall rating. Since our use of BBSC to construct continuous manifold spaces is relatively novel, some reviewers may have initially underestimated its significance. Fortunately, during the discussion phase, we were able to clarify the novelty, importance, and implications of our work extensively.

Finally, we will incorporate all responses into the final revision and release the code upon publication. We deeply appreciate the opportunity for publication and thank all involved for their efforts.

---

### Decision · Program_Chairs · 2025-09-17

**Decision:**

Accept (poster)

**Comment:**

This paper introduces a novel framework for modeling tubular branching structures using BBSC manifolds. While one reviewer raised concerns about presentation quality and mathematical clarity, the majority view the contribution as interesting, technically sound, and potentially impactful. With novelty and relevance to the community, the overall balance of reviews supports acceptance.

I would strongly encourage the authors to incorporate the new results, as well as clarifications, especially the concerns of reviewer G1qr, into the revised version.